# Calculating the vertical column density of $O_4$ during daytime from surface values of pressure, temperature and relative humidity

Steffen Beirle[1], Christian Borger[1], Steffen Dörner[1], Vinod Kumar[1], and Thomas Wagner[1]

[1]Max-Planck-Institut für Chemie

**Correspondence:** Steffen Beirle (steffen.beirle@mpic.de)

**Abstract.** We present a formalism that relates the vertical column density (VCD) of the oxygen collision complex $O_2$-$O_2$ (denoted as $O_4$ below) to surface ($2\,\mathrm{m}$) values of temperature and pressure, based on physical laws. In addition, we propose an empirical modification which also accounts for surface relative humidity (RH). This allows for simple and quick calculation of the $O_4$ VCD without the need for constructing full vertical profiles. The parameterization reproduces the true $O_4$ VCD, as derived from vertically integrated profiles, within $-0.7\,\% \pm 1.2\,\%$ (mean±SD) for WRF simulations around Germany, $0.2\,\% \pm 1.8\,\%$ for global reanalysis data (ERA5), and $-0.3\,\% \pm 1.4\,\%$ for GRUAN radiosonde measurements around the world. When applied to measured surface values, uncertainties of $1\,\mathrm{K}$, $1\,\mathrm{hPa}$, and $16\,\%$ for temperature, pressure, and RH correspond to relative uncertainties of the $O_4$ VCD of $0.3\,\%$, $0.2\,\%$, and $1\,\%$, respectively. The proposed parameterization thus provides a simple and accurate formula for the calculation of the $O_4$ VCD which is expected to be useful in particular for MAX-DOAS applications.

## 1 Introduction

In the atmosphere, two oxygen molecules can build collision pairs and dimers, which are often denoted as $O_4$ (Greenblatt et al., 1990; Thalman and Volkamer, 2013, and references therein). $O_4$ has absorption bands in the UV/visible spectral range, thus $O_4$ can be retrieved from atmospheric absorption spectra, e.g. by applying Differential Optical Absorption Spectroscopy (DOAS) (Platt and Stutz, 2008). Measurements of the $O_4$ absorption pattern in scattered light provide information about light path distributions in the atmosphere, for instance allowing to investigate light path length increase within clouds (Wagner et al., 1998) or the retrieval of cloud heights from satellite measurements (Acarreta et al., 2004; Veefkind et al., 2016).

For Multi-Axis (MAX) DOAS, i.e. ground-based instruments measuring scattered light at different elevation angles, $O_4$ measurements provide information on vertical profiles of aerosol extinction (Heckel et al., 2005). Prerequisite for MAX-DOAS profile inversions is knowledge about the $O_4$ vertical column density (VCD) which provides the link between the measured slant column densities (SCDs) at different viewing angles and the forward modelled SCDs based on radiative transfer calculations. Thus, a wrong input of the $O_4$ VCD directly affects the resulting aerosol profiles. For the profile inversion algorithm MAPA (Beirle et al., 2019) applied to measurements taken during the CINDI-2 campaign (Kreher et al., 2020), for instance, a change of the input $O_4$ VCD of $2\,\%$, $3\,\%$, $5\,\%$, or $10\,\%$ causes changes of the resulting median aerosol optical depth of $6\,\%$, $8\,\%$, $13\,\%$, or $20\,\%$, respectively. Thus, for MAX-DOAS profile inversions, the $O_4$ VCD should be determined

with accuracy and precision better than about $3\%$, which limits the impact on resulting AODs to below $10\%$ and leaves other sources of uncertainty, i.e. the spectral analysis ($\approx 5\%$) as well as radiative transfer modeling ($\approx 4\%$) (see Wagner et al., 2021, Table 3 therein) as the limiting factors.

The $O_4$ VCD can be calculated by vertical integration of the $O_2$ number density profile squared. This requires knowledge of vertical profiles of temperature, pressure, and humidity, e.g. as derived from radiosonde measurements or meteorological models. However, radiosonde measurements are only available for few stations and do not provide continuous temporal coverage, while modelled profiles might not be available in some cases (e.g. during measurement campaigns in remote regions and poor internet connection; for these cases, profiles from a climatology might be used as fallback option), or might not reflect the conditions at the measurement site appropriately, in particular in mountainous terrain not resolved by the model.

Measurements of surface air (at $2\,\mathrm{m}$) temperature, pressure, and humidity, on the other hand, are routinely performed by meteorological stations, and could be added to any MAX-DOAS measurement site with relatively low costs and efforts. Wagner et al. (2019) proposed to construct full temperature and pressure profiles from the respective surface values by assuming (a) a constant lapse rate of $-6.5\,\mathrm{K}\,\mathrm{km}^{-1}$ from ground up to $12\,\mathrm{km}$, and constant temperature above, and (b) applying the barometric formula.[1] Wagner et al. (2019) estimate the uncertainty of the calculated $O_4$ VCD to $3\%$ and list the diurnal variation of the surface temperature and the limited representativeness of the surface temperature for the temperature profile above the boundary layer as the main source of uncertainty.

The method proposed by Wagner et al. (2019) reproduces the true $O_4$ VCD within about $2\%$ (bias) $\pm2\%$ standard deviation (SD) globally when compared to ECMWF profiles. Locally, however, large deviations up to $7\%$ could be found, as shown in the current study, which is mainly caused by the assumption of a fixed lapse rate of $-6.5\,\mathrm{K}\,\mathrm{km}^{-1}$: While this value reflects typical continental conditions quite well, it is not appropriate in particular over deserts, where lapse rates are stronger (closer to the dry adiabatic lapse rate), and parts of the oceans with weaker (i.e. closer to zero) lapse rates due to condensation.

In this paper we present a simpler approach for the calculation of $O_4$ VCD just from surface values of temperature and pressure and an a-priori lapse rate based on physical laws, without the need of constructing full profiles. In addition, we provide an empirical parameterization involving surface relative humidity that also accounts for variations of the atmospheric lapse rate. This allows for simple and quick calculation of the $O_4$ VCD with high accuracy and precision just from surface measurements of temperature, pressure, and relative humidity.

The manuscript derives the formalism of the parameterizations of the $O_4$ VCD in Sect. 2. In Sect. 3, the datasets used for illustration and quantification of uncertainties are introduced, followed by applications of the $O_4$ parameterizations in Sect. 4. Important aspects like comparison to standard methods used for the calculation of the $O_4$ VCD, the impacts of temperature inversions, surface altitude, or diurnal cycles, and the accuracy and precision of the proposed parameterizations are discussed in Sect. 5, followed by conclusions.

---

[1]Note that, for this approach, as well as for the parameterizations presented in this study, temperature inversions are problematic. As MAX-DOAS applications require daylight, however, night-time inversion layers are irrelevant for this study. The remaining temperature inversions at daytime, mostly occurring in early morning hours and over cold water and ice surfaces, will be discussed in Sect. 5.2.

## 2 Formalism

In this section, we provide the formalism for the calculation of $O_4$ VCDs from surface values of pressure, temperature, and relative humidity.

### 2.1 Notation

Basic quantities of the derivation below are (a) the number density $n$, and (b) the vertical column density (VCD) $V$, i.e. the vertically integrated number density.

The $O_4$ number density is just defined as the $O_2$ number density squared. Consequently, the $O_4$ number density has the unit $\text{molecules}^2 \, \text{cm}^{-6}$, and the $O_4$ VCD has the unit $\text{molecules}^2 \, \text{cm}^{-5}$. This matches the common procedure in the DOAS community; the $O_4$ cross section is given in $\text{cm}^5 \, \text{molecules}^{-2}$ accordingly (Greenblatt et al., 1990; Thalman and Volkamer, 2013).

Pressure is denoted by $p$, temperature by $T$, and the altitude above sea level by $z$, while altitude above ground level is denoted by $z'$. For relative humidity, RH is used in the text as well as in formulas. Surface values are indicated by the subscript "0". Quantities related to $O_2$ and $O_4$ are indicated by a respective subscript. For a full list of quantities and symbols see Tables 1 and 2.

**Table 1.** Variables used in this study. A subscript of 0 indicates surface values for $n$, $p$, $T$, $z$, or RH.

| Quantity | Acronym | Symbol | Unit |
|---|---|---|---|
| Number density | - | $n_{O_2}$ | $\text{molecules} \, \text{cm}^{-3}$ |
| | | $n_{O_4}$ | $\text{molecules}^2 \, \text{cm}^{-6}$ |
| Vertical column density | VCD | $V_{O_2}$ | $\text{molecules} \, \text{cm}^{-2}$ |
| | | $V_{O_4}$ | $\text{molecules}^2 \, \text{cm}^{-5}$ |
| Pressure | - | $p$ | hPa |
| Temperature | - | $T$ | K |
| Altitude above sea level | - | $z$ | m |
| Altitude above surface | - | $z'$ | m |
| Effective height | - | $h$ | m |
| Scale height | - | $H$ | m |
| Relative humidity | RH | RH | |
| Lapse rate | - | $\Gamma$ | $\text{K} \, \text{km}^{-1}$ |
| Relative deviation from true $O_4$ VCD | - | $\delta$ | % |
| Top of atmosphere[a] | TOA | $z_{\text{TOA}}$ | m |
| Total column water vapor | TCWV | $V_{\text{H}_2\text{O}}$ | $\text{molecules} \, \text{cm}^{-2}$ |

[a] here: highest available/possible profile layer.

**Table 2.** Constants used in this study. Numbers are listed with 6 digits.

| Quantity | Symbol | Value | Unit |
|---|---|---|---|
| Gravitational acceleration on Earth | $g$ | $9.80665^{a, b}$ | $\mathrm{m\,s^{-2}}$ |
| Molar mass of dry air | $M$ | $0.0289655^{a}$ | $\mathrm{kg\,mol^{-1}}$ |
| Universal gas constant | $R$ | $8.31446^{a}$ | $\mathrm{J\,K^{-1}\,mol^{-1}}$ |
| $O_2$ volume mixing ratio in dry air | $\nu_{O_2}$ | $0.209392^{c}$ | |
| Combined constants (Eq. 11) | $C$ | $6.73266 \cdot 10^{39}$ | $\mathrm{K\,hPa^{-2}\,molecules^2\,cm^{-5}}$ |

[a] from the Python module MetPy (May et al., 2021).
[b] latitudinal variations of $g$ are below $\pm 0.27\,\%$ and are neglected in this study.
[c] from Tohjima et al. (2005)

## 2.2 General approach

The VCD $V$ is the vertically integrated number density $n$:

$$V = \int_{z_0}^{\infty} n(z)\,dz \tag{1}$$

This integral can be re-written as

$$V = n_0 \cdot h, \tag{2}$$

with

$$h = \int_{z_0}^{\infty} \frac{n(z)}{n_0}\,dz \tag{3}$$

This effective height $h$ can be understood as the height of the gas column if the gas would be in a homogenous box under surface conditions $p_0$ and $T_0$. Note that the effective height equals the scale height $H$ only in case of exponential profiles,
i.e. an isothermal atmosphere (see Appendix A).

Thus, the VCDs for $O_2$ and $O_4$ can be written as

$$V_{O_2} = n_{O_2,0} \cdot h_{O_2} \tag{4}$$

and

$$V_{O_4} = n_{O_4,0} \cdot h_{O_4} = n_{O_2,0}^2 \cdot h_{O_4} \tag{5}$$

Re-arranging Eq. 4 for $n_{O_2,0}$ and replacing one $n_{O_2,0}$ term in Eq. 5 yields

$$V_{O_4} = V_{O_2} \cdot n_{O_2,0} \cdot \frac{h_{O_4}}{h_{O_2}} \tag{6}$$

Hence the $O_4$ VCD can be expressed as the product of the $O_2$ VCD, the $O_2$ surface number density, and the ratio of effective heights of $O_2$ and $O_4$ profiles. So far, no simplifications or approximations were made, thus Eq. 6 holds for any atmospheric conditions.

## 2.3 $O_4$ VCD as a function of surface pressure, surface temperature, and lapse rate

Based on Eq. 6, the $O_4$ VCD can be related to surface pressure, surface temperature, and lapse rate, if some further assumptions are made:

1. Assuming a hydrostatic atmosphere, the surface pressure is just the gravitational force per area of the total air column. Thus, the $O_2$ VCD is directly related to the surface pressure:

$$V_{O_2} = \frac{\nu_{O_2}}{g \cdot M} \cdot p_0, \tag{7}$$

with $\nu_{O_2}$ being the volume mixing ratio of $O_2$ in dry air, $g$ being the gravitational acceleration on Earth, and $M$ being the molar mass of dry air.

2. According to the ideal gas law for dry air, the surface number density of $O_2$ can be expressed as

$$n_{O_2,0} = \frac{\nu_{O_2}}{R} \cdot \frac{p_0}{T_0}, \tag{8}$$

with the universal gas constant $R$.

3. The ratio of effective heights for $O_2$ and $O_4$ depends on the actual profile shape for $O_2$. For some specific cases, the integral in Eq. 3 can be solved analytically (see Appendix A for details):

   – For an isothermal atmosphere, i.e. an exponential profile of $n_{O_2}$, the ratio $\frac{h_{O_2}}{h_{O_4}}$ is just 2 (Eq. A2).
   – For a constant lapse rate $\Gamma$, the ratio becomes $2 + \frac{R}{g \cdot M}\Gamma$ (Eq. A10).
   – For real atmospheric conditions, where the lapse rate varies with altitude, the ratio of effective heights can be still described by Eq. A10 if an *effective* lapse rate is considered:

$$\Gamma_{\text{eff}} := \left( \frac{h_{O_2}}{h_{O_4}} - 2 \right) \cdot \frac{g \cdot M}{R} \tag{9}$$

   For a given atmospheric profile, the effective lapse rate is thus defined as the lapse rate of a polytropic atmosphere of the same $O_4$ VCD.

Replacing the terms in Eq. 6 by Eq. 7, Eq. 8 and Eq. A10 yields

$$V_{O_4,\Gamma} = \frac{\nu_{O_2}^2}{R \cdot g \cdot M} \bigg/ \left( 2 + \frac{R}{g \cdot M}\Gamma \right) \cdot \frac{p_0^2}{T_0}$$

$$= \frac{C}{2 + \frac{R}{g \cdot M}\Gamma} \cdot \frac{p_0^2}{T_0} \tag{10}$$

with

$$C = \frac{\nu_{O_2}^2}{R \cdot g \cdot M} \tag{11}$$

combining the constant factors. Thus with the assumptions specified above, the $O_4$ VCD is proportional to $p_0^2/T_0$, with the lapse rate $\Gamma$ determining the slope.

## 2.4 Calculation of the "true" $O_4$ VCD

In order to evaluate the performance of the parameterizations for the $O_4$ VCD, we also calculate the "true" $O_4$ VCD, which is derived by

(a) calculating the profile of $n_{O_2}$ from profiles of $T$, $p$ and RH. In this step, the effect of humidity is explicitly accounted for by subtracting the water vapor pressure before calculating $n_{O_2}$ based on the ideal gas law.

(b) performing the numerical integration (using Simpson's rule) of $n_{O_2}^2$ from surface to top of atmosphere (TOA).

The integration has to be performed up to sufficiently high altitudes (Wagner et al. (2019) recommend $z_{TOA} \geq 30\,\text{km}$) as otherwise the integrated VCD would be biased low due to the missing column above. As not all datasets considered below cover this altitude range, we estimate and correct for the missing $O_4$ column above the highest profile level by applying Eq. 10 for the highest available layer, assuming a lapse rate of zero above. Note that the temperature increase in the upper stratosphere is not relevant here as the contribution to the $O_4$ VCD above 30 km is negligible. Thus, the "true" $O_4$ VCD is calculated as

$$V_{O_4,\text{true}} = \int_{z_0}^{z_{TOA}} n_{O_2}^2(z)dz + \frac{C}{2} \cdot \frac{p_{TOA}^2}{T_{TOA}} \tag{12}$$

For $z_{TOA}$ of 20 km, the correction term is of the order of $0.3\,\%$ of the total $O_4$ column.

## 2.5 $O_4$ VCD as a function of surface pressure, surface temperature, and surface relative humidity

Equation 10 might be applied for a common lapse rate, like $-6.5\,\text{K}\,\text{km}^{-1}$ as proposed in Wagner et al. (2019). This works generally well over most continental regions. However, large deviations have to be expected for regions with different lapse rates, in particular over deserts, where lapse rates are typically stronger (more negative, i.e. close to the dry adiabatic lapse rate). Over parts of the ocean, on the other hand, lapse rates are weaker (closer to zero).

In order to modify Eq. 10 such that it can be applied globally, but keep it still a simple function of surface measurements, we make use of the relation between the effective lapse rate and the RH at ground:

– For ascending air masses, $RH_0$ determines the altitude at which condensation takes place. This relation is directly reflected in the calculation of the lifted condensation level (LCL) as function of $RH_0$ (Lawrence, 2005; Romps, 2017). Thus, the lower $RH_0$, the higher the altitude range above ground where dry adiabatic lapse rates apply.

– For descending air masses (in particular the large-scale subsidence over tropical deserts), no condensation takes place and dry adiabatic lapse rates apply.

In both cases, low relative humidity at ground is associated with lapse rates closer to the dry adiabatic lapse rate.

Real atmospheric profiles are of course more complex than these simplified scenarios, in particular due to advection, but still, a correlation between $RH_0$ and effective lapse rates is expected. We thus parameterize the effective lapse rate by the relative humidity at ground via a linear function:

$$\Gamma_{\text{eff}} = \alpha + \beta \cdot RH_0 \tag{13}$$

Replacing this in Eq. 10 results in $V_{O_4}$ becoming a function of surface values for pressure, temperature, and relative humidity:

$$\begin{aligned} V_{O_4,RH} &= \frac{C}{2 + \frac{R}{g \cdot M} \cdot (\alpha + \beta \cdot RH_0)} \cdot \frac{p_0^2}{T_0} \\ &= \frac{C}{a + b \cdot RH_0} \cdot \frac{p_0^2}{T_0}, \end{aligned} \tag{14}$$

where the parameters $a$ and $b$ are linked to $\alpha$ and $\beta$ from Eq. 13 via

$$\alpha = (a - 2) \cdot \frac{g \cdot M}{R} \tag{15}$$

and

$$\beta = \frac{g \cdot M}{R} \cdot b. \tag{16}$$

The parameters $a$ and $b$ can then be determined by a linear least squares fit by comparing $V_{O_4,RH}$ to $V_{O_4,\text{true}}$:

$$V_{O_4,RH} = \frac{C}{a + b \cdot RH_0} \cdot \frac{p_0^2}{T_0} \stackrel{!}{=} V_{O_4,\text{true}}, \tag{17}$$

thus

$$a + b \cdot RH_0 \stackrel{!}{=} \frac{C}{V_{O_4,\text{true}}} \cdot \frac{p_0^2}{T_0} =: Q. \tag{18}$$

$a$ and $b$ will be derived in Sect. 4.2 based on true $O_4$ VCDs calculated from ECMWF profiles. This empirical approach also, at least partly, corrects for effects neglected in the derivation of Eq. 10, i.e. ignoring the tropopause in the calculation of the ratio of effective heights (App. A3), and applying the ideal gas law for dry air in Sect. 2.3.

From $a$ and $b$, also the corresponding effective lapse rate for a given $RH_0$ can then be calculated with Equations 13, 15 and 16. This lapse rate allows to construct full atmospheric profiles of $T$ and $p$ (applying the barometric formula for polytropic atmosphere) from surface measurements when needed, in particular for MAX-DOAS inversions based on optimal estimation. As humidity effects are already accounted for in the determination of $a$ and $b$, no further correction for humidity should be applied in this case.

## 2.6 Comparison of parameterized to "true" $O_4$ VCD

In order to assess accuracy and precision of the proposed calculation of the $O_4$ VCD from surface measurements of $T_0$, $p_0$ and $\Gamma$ (Eq. 10) or $RH_0$ (Eq. 14), we define the relative deviation $\delta$ of a derived $O_4$ VCDs to the true value:

$$\delta_x = \frac{V_{O_4,x} - V_{O_4,\text{true}}}{V_{O_4,\text{true}}} \tag{19}$$

The index $x$ indicates the $O_4$ VCD dataset and is "Γ" for $V_{O_4,\Gamma}$ and "RH" for $V_{O_4,RH}$.

## 3  Datasets

For illustration as well as for the quantification of uncertainties, we apply the derived formalism to different atmospheric datasets:

1. Global model data, in order to check for the performance of the parameterizations globally, covering the full range of the relevant parameter space for surface values of pressure, temperature, humidity, and altitude.

2. Regional model data with high spatial resolution, which are also compared to surface stations and allow to investigate the impact of diurnal cycles.

3. Balloon-borne radiosonde measurements, in order to apply the formalism to high-resolved profile measurements.

Nighttime profiles of $T$ can be considerably different from daytime profiles, in particular in case of temperature inversions (i.e. positive lapse rates) often occurring within the nocturnal boundary layer. For MAX-DOAS measurements, however, nighttime profiles are irrelevant. Thus, we consider all atmospheric datasets for daytime conditions only. This is done by selecting data with an solar zenith angle (SZA) below $85°$.

### 3.1  Global model (ECMWF)

We use global model data as provided by the European Centre for Medium-Range Weather Forecasts (ECMWF) for two purposes:

– In order to investigate global patterns, we use ERA5 reanalysis data (Hersbach et al., 2020) truncated at wavenumber 639 on the Gaussian grid N320, corresponding to $\approx0.3°$ resolution. Model output is provided hourly. Here, we focus on ERA5 data for four selected days, i.e. the 18 March, 18 June, 18 September and 18 December 2018, covering the full globe for all seasons. As the regular latitude-longitude grid over-represents high latitudes, we only consider the fraction of cos(lat) pixels for each latitude for the calculation of histograms, correlation coefficients, means and standard deviations.

– For comparison with the standard approach for the calculation of the $O_4$ VCD that was used in MAPA so far, we use ERA-Interim reanalysis data truncated at wavenumber 255, corresponding to $\approx0.7°$ resolution, which was preprocessed to a dataset with 6 hourly model output (0:00, 6:00, 12:00, 18:00 UTC) interpolated to a regular horizontal grid with a resolution of $1°$. This dataset is denoted as ERA_$I_{daily}$ below.

In addition, we make use of a monthly climatology of atmospheric profiles (ERA_$I_{clim}$) based on the same ERA-Interim data, which was constructed as back-up solution recommended within the FRM4DOAS project in case of no other profile information being available.

## 3.2 Regional model (WRF-Chem) and surface measurements (DWD)

We use the Weather Research and Forecasting (WRF) model version 4.2 (Skamarock et al., 2019) for high resolution ($3 \times 3 \, \text{km}^2$) simulations of meteorological parameters (including $T$, $p$ and RH) around Germany for May and June 2018. Further details on the WRF model set-up are provided in Appendix B1.1.

WRF simulations of surface values are also compared to surface measurements performed by Germany's National Meteorological Service (Deutscher Wetterdienst, DWD). For further details see Appendix B1.2 and B1.3.

## 3.3 Radiosonde measurements (GRUAN)

The Global Climate Observing System (GCOS) Reference Upper-Air Network (GRUAN) is an international reference observing network of sites measuring essential climate variables above Earth's surface (Sommer et al., 2012; Bodeker et al., 2016). Atmospheric profiles of temperature, pressure, and humidity are measured by regular balloon soundings equipped with radiosondes and water vapor measurements (Dirksen et al., 2014). Here we use the RS92 GRUAN Data Product Version 2 (RS92-GDP.2), focusing on certified stations. Vertical profiles and surface values of pressure, temperature and relative humidity are taken directly from the level-2 files for each launch. Further information on the GRUAN stations used in this study are provided in Appendix B2.

## 4 Application to atmospheric datasets

In this section, we apply the parameterizations of the $O_4$ VCD derived in Sect. 2 to modeled and measured atmospheric datasets, and assess the accuracy and precision of the different parameterizations by comparison to the true $O_4$ VCD (Sect. 2.4). We first present $O_4$ VCDs based on an a priori lapse rate in Sect. 4.1, discuss the relation between effective lapse rate and surface humidity in Sect. 4.2, and finally present $O_4$ VCDs based on $RH_0$ in Sect. 4.3.

### 4.1 $O_4$ VCD as a function of $p_0$, $T_0$, and lapse rate $\Gamma$

According to Eq. 10, the $O_4$ VCD is proportional to $p_0^2/T_0$, with the lapse rate $\Gamma$ determining the slope. We illustrate this correlation for the investigated datasets in Fig. 1.

$V_{O_4, \, \text{true}}$ varies considerably for all datasets, where the low values are caused by mountains due to reduced pressure, while the very high values for ERA5 and GRUAN are caused by cold temperatures in polar regions. The variability of $V_{O_4, \, \text{true}}$ is well reflected in $p_0^2/T_0$, and very good correlation between both quantities is found, with most data points in accordance to plausible lapse rates in the range of $-4$ to $-6.5 \, \text{K km}^{-1}$. For the WRF simulations for Germany, most data points are matching to a lapse rate close to $-6.5 \, \text{K km}^{-1}$. ERA5 and GRUAN data show higher variability in slopes, as they also cover a wider range of atmospheric conditions.

Wagner et al. (2019) proposed to determine the $O_4$ VCD based on vertical profiles of $T$ and $p$ constructed from the respective surface values by assuming a constant tropospheric lapse rate of $-6.5 \, \text{K km}^{-1}$. We can use Eq. 10 for the same purpose, but

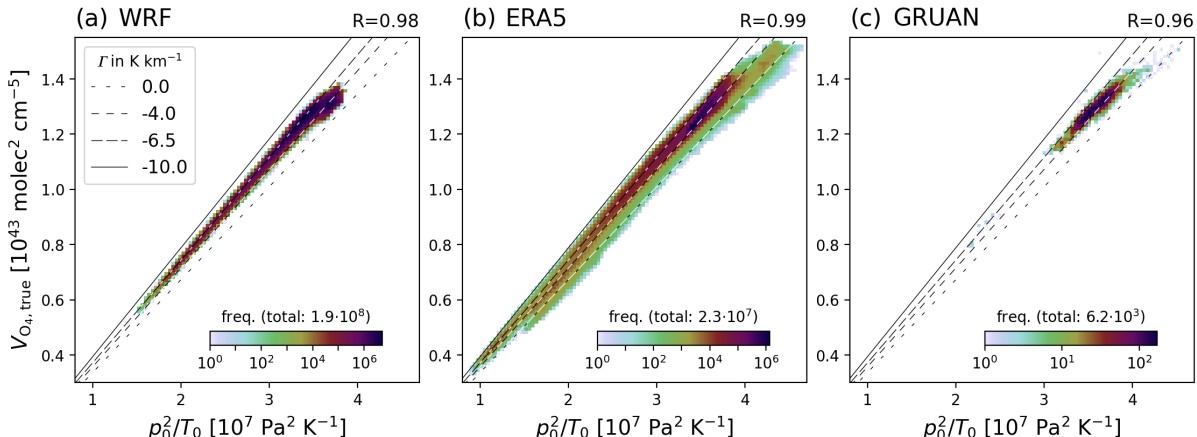

**Figure 1.** Density plots of the relation between the $O_4$ VCD and $p_0^2/T_0$ for (a) WRF data for May to June 2018 from 7:00 to 17:00 UTC, (b) ERA5 data for the 4 selected days in 2018, and (c) all available GRUAN profiles. Frequency per pixel is color coded, with a binning of 100 pixels for both $x$ and $y$ axis, as in all 2D frequency distributions shown below. Respective correlation coefficients are provided in the top right of each panel. Lines display the expected dependency according to Eq. 10 for different lapse rates. Low values correspond to high altitude sites with low surface pressure; for GRUAN, only few of such stations are available (Boulder and La Reunion at 1.7 km and 2.2 km altitude, respectively). The very high values for ERA5 and GRUAN are observed for polar regions with cold temperatures.

without the need for constructing full vertical profiles. Figures 2 and 3 display maps of the deviation $\delta_\Gamma$ between parameterized and true $O_4$ VCD for WRF and ERA5, respectively, assuming a constant lapse rate of $-6.5\,\mathrm{K\,km^{-1}}$. Results for additional days for ERA5 are shown in Appendix C.

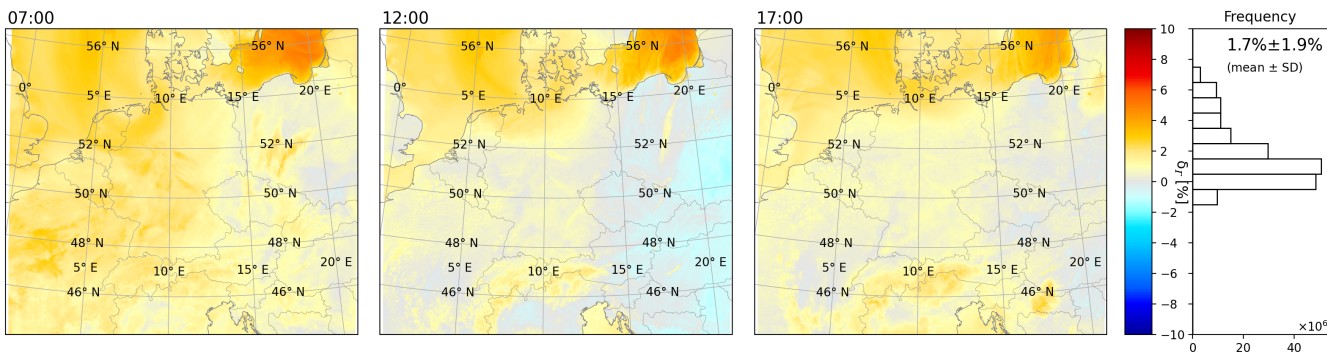

**Figure 2.** Deviation $\delta_\Gamma$ according to Eq. 19 for WRF simulations at 7:00, 12:00, and 17:00 UTC on 1 May 2018. On the right, the frequency distribution of $\delta_\Gamma$ and its mean and SD are given for the WRF simulation period from May to June 2018.

Generally, good agreement between $V_{O_4,\,\Gamma}$ and $V_{O_4,\,\text{true}}$ is found (Fig. 2): on average, $\delta_\Gamma$ is 1.7 %, i.e. $V_{O_4,\,\Gamma}$ are higher than $V_{O_4,\,\text{true}}$ by 1.7 %. Over land around noon, $\delta_\Gamma$ is close to 0. Over ocean, however, $\delta_\Gamma$ is higher (about 3 % up to 7 %).

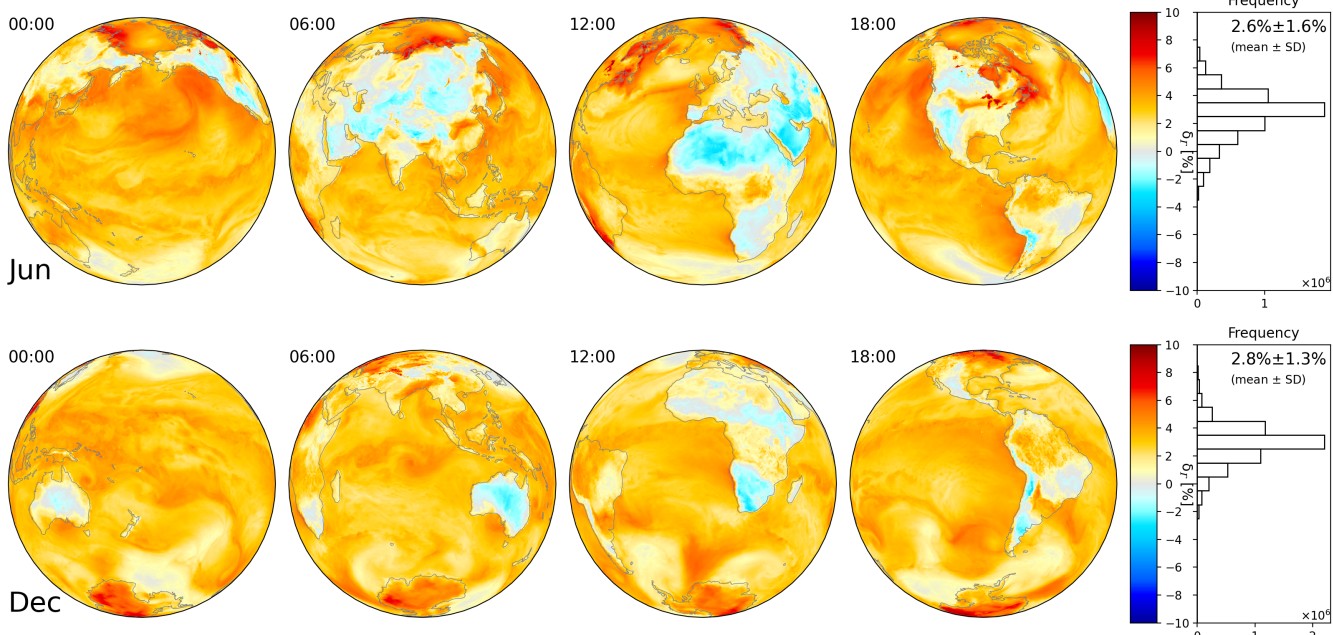

**Figure 3.** Deviation $\delta_\Gamma$ according to Eq. 19 for ERA5 at 0:00, 6:00, 12:00 and 18:00 UTC on 18 June (top) and 18 December (bottom) 2018. The projection focuses on daytime for each timestep. On the right, the frequency distribution and mean and SD are given for all hourly outputs of the respective day. Results for 18 March and 18 September 2018 are shown in Fig. C1.

For ERA5 data on 18 June 2018, $\delta_\Gamma$ over Germany is close to 0 as well (Fig. 3). On global scale, however, only moderate agreement is found between $V_{O_4, \Gamma}$ and $V_{O_4, \text{true}}$, with a mean deviation of 2.6 %. High values for $\delta_\Gamma$ are found generally over ocean. In particular over cold water surfaces, like the West coast of North and South America, the Hudson Bay, or the

235 Great lakes, $\delta_\Gamma$ is very high (up to 7 %). This is related to temperature inversions close to ground: due to the too low surface temperatures (as compared to a polytropic atmosphere with the same $O_4$ VCD), the $O_4$ VCD calculated from Eq. 10 is biased high. This will be discussed in detail in Sect. 5.2. Over continents, $\delta_\Gamma$ is lower, and generally close to 0, except over deserts, where negative values are observed.

### 4.2 Effective lapse rate and relative humidity at ground

Figure 4 displays the effective lapse rate, as defined in Eq. 9, for ERA5 data from 18 June 2018, clearly showing that the general patterns of systematic deviations seen in Fig. 3 are mainly caused by the simple assumption of a globally constant lapse rate in the calculation of $\delta_\Gamma$.

In Fig. 5 (a), the effective lapse rate is compared to the actual lapse rate between ground and 5 km altitude above ground, revealing a correlation of 0.83. Figure 5 (b) displays the relation between relative humidity at ground and the effective lapse rate (R=0.59). Assuming a linear relation between $\Gamma_{\text{eff}}$ and $RH_0$ in Eq. 13 results in a linear relation between $RH_0$ and the

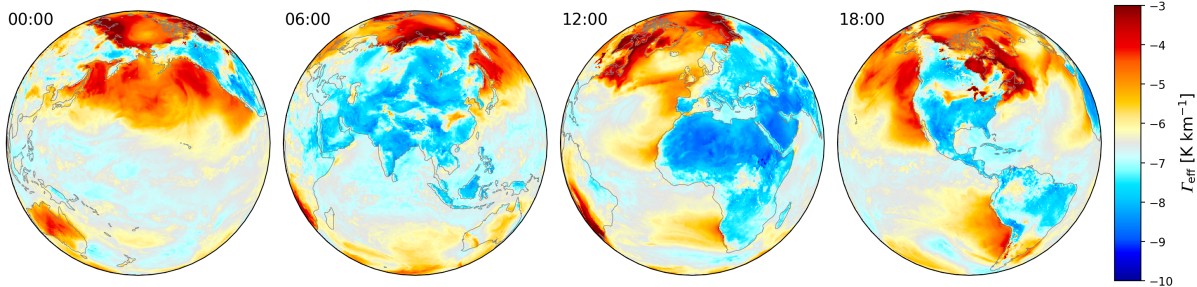

**Figure 4.** Effective lapse rate $\Gamma_{\mathrm{eff}}$ as defined in Eq. 9 for ERA5 profiles on 18 June 2018.

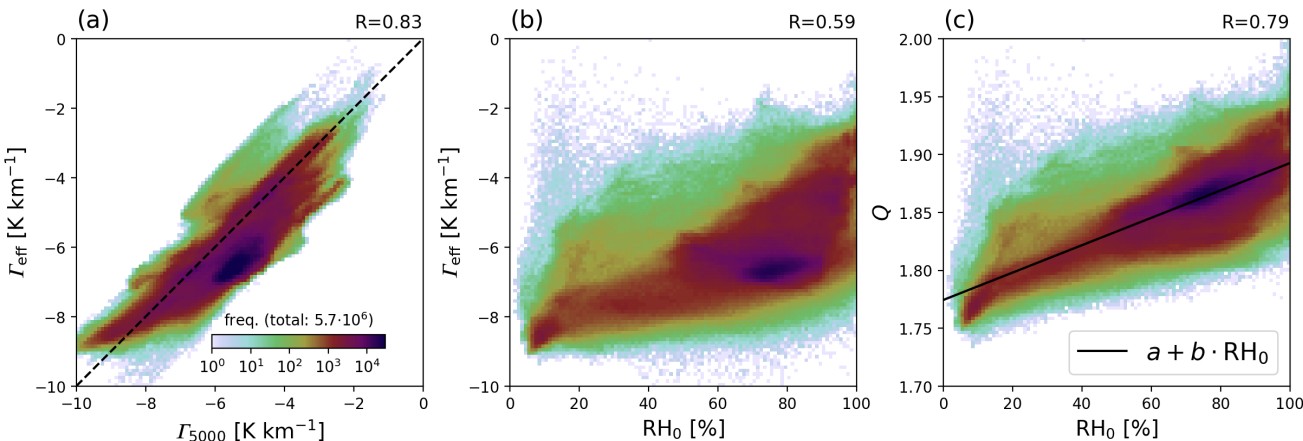

**Figure 5.** Relations between lapse rate, $RH_0$ and ratio $Q$ for ERA5 data on 18 June 2018. (a) Relation between the actual lapse rate (calculated from the temperature difference between ground and 5 km altitude) and the effective lapse rate. (b) Relation between $RH_0$ and effective lapse rate. (c) Relation between $RH_0$ and the ratio $Q$ as defined in Eq. 18, where the black line shows the linear fit $a + b \cdot RH_0$.

ratio $Q$ (Eq. 18). This is displayed in Fig. 5 (c). Interestingly, the correlation (R=0.79) is far better than in (b) (R=0.59). This indicates that the parameterization based on $RH_0$ at least partly corrects for other simplifications made in the formalism in Sect. 2.3, in particular the neglect of humidity in the ideal gas law.

We use ERA5 data from 18 June 2018 to determine the parameters $a$ and $b$ in Eq. 14 by applying a linear least squares fit to the
data presented in Fig. 5 (c), as shown by the black line. Fitted parameters are $a = 1.77434 \pm 0.00003$ and $b = 0.11821 \pm 0.00004$ (for $RH_0$ in absolute numbers, i.e. 0.5 for 50% RH). The corresponding parameterisation of the effective lapse rate (Eq. 13) is $\Gamma_{\mathrm{eff}} = (-7.709 + 4.038 \cdot RH_0)$ K km$^{-1}$, which yields -7.709, -5.690 and -3.671 K km$^{-1}$ for $RH_0$ of 0%, 50%, and 100%, respectively.

### 4.3 $O_4$ VCD as function of $p_0$, $T_0$, and $RH_0$

With Eq. 14, an empirical parameterization of the $O_4$ VCD was derived based on surface values of temperature, pressure, and relative humidity. We applied this parameterization to all investigated datasets. Figures 6 and 7 display $\delta_{RH}$ for WRF and ERA5, respectively. GRUAN results are shown in Fig. 8.

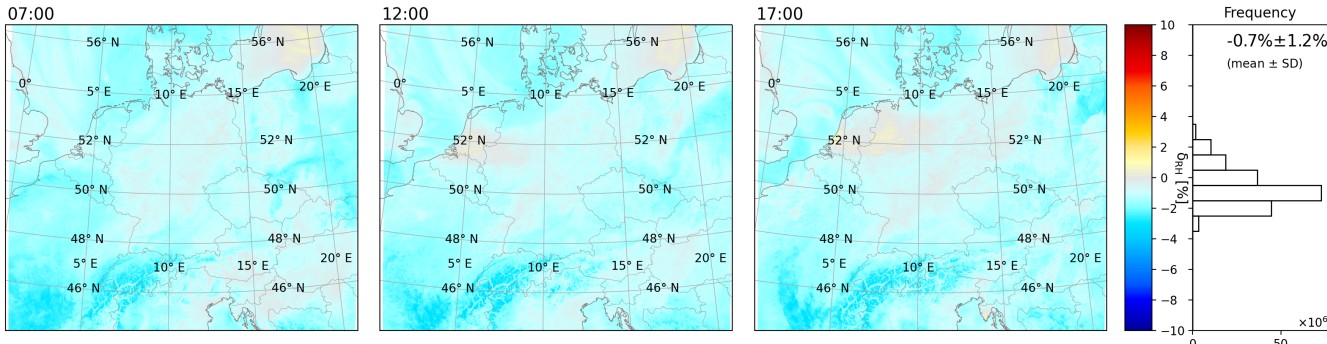

**Figure 6.** Deviation $\delta_{RH}$ according to Eq. 19 for WRF simulations at 7:00, 12:00, and 17:00 UTC on 1 May 2018. On the right, the frequency distribution and mean and SD are given for the WRF simulation period from May to June.

For the WRF simulations, $\delta_\Gamma$ was already quite close to 0 (mean $\delta_\Gamma = 1.6\%$, see Fig. 2). $\delta_{RH}$ (Fig. 6) is closer to 0, but now showing a slight negative bias (mean $\delta_{RH} = -0.7\%$). Variability has reduced considerably (SD of $\delta_{RH}$ is $1.2\%$, compared to $1.9\%$ for $\delta_\Gamma$). $\delta_{RH}$ shows a weaker land-ocean contrast. Over the Alps, $\delta_{RH}$ is biased low (down to $-3\%$).

For ERA5, the parameterization involving RH is a substantial improvement compared to the results for $\delta_\Gamma$. The large difference between land (in particular deserts) and oceans seen in $\delta_\Gamma$ (Fig. 3) is strongly reduced for $\delta_{RH}$ (Fig. 7). For 18 June 2018, the mean of $\delta_{RH} \equiv 0.0\%$ is of course a consequence of the fit optimizing $a$ and $b$ which is based on the same ERA5 dataset. But also for 18 December 2018, the mean deviation is close to zero. The SD reduces from $1.6\%$ for $\delta_\Gamma$ to $1.0\%$ for $\delta_{RH}$ on 18 June 2018, and from $1.3\%$ to $1.2\%$ on 18 December 2018. Applying Eq. 14 to ECWMF data from other months yields similar results, as shown in Fig. C2, with largest deviations of $0.2 \pm 1.8\%$ observed for 18 March 2018.

Remaining systematic deviations in the maps of $\delta_{RH}$ are due to

– weather, for instance associated with low pressure or frontal systems. This reflects the simplifying assumptions made, in particular assuming hydrostatic conditions in Sect. 2. Note, however, that MAX-DOAS retrievals are usually not considered for weather conditions associated with rain and clouds.

– cold surfaces causing temperature inversions, as discussed in more detail in Sect. 5.2.

– mountains, which tend to show systematic deviations $\delta_{RH}$ that are mostly negative (e.g. over the Andes or the Himalayas). For further discussion see Sect. 5.4.

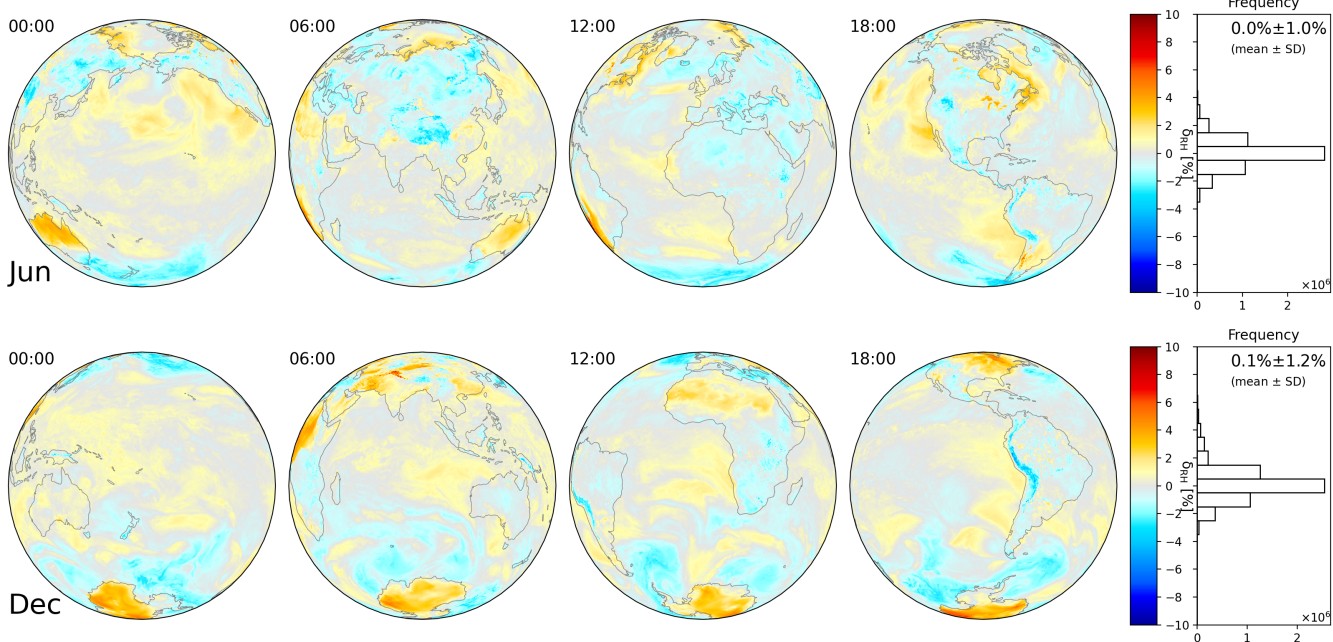

**Figure 7.** Deviation $\delta_{RH}$ according to Eq. 19 for ERA5 at 0:00, 6:00, 12:00 and 18:00 UTC on 18 June (top) and 18 December (bottom) 2018. The projection focuses on daytime for each timestep. On the right, the frequency distribution and mean and SD are given for all hourly outputs of the respective day. Note that the mean deviation on 18 June 2018 is 0 by construction, as the parameters $a$ and $b$ (Eq. 14) are optimized for this day. Results for 18 March and 18 September 2018 are shown in Fig. C2.

- some patterns of enhanced $\delta_{RH}$ at the Northern, Southern, or Western edge of the maps for ERA5, corresponding to polar regions as well as sampling times shortly after sunrise (e.g. over South Africa at 6:00 UTC).

So far, the formalism derived in Sect. 2 was applied to data from meteorological models. Now we test it for *measured* profiles from radiosondes as well. Application of Eq. 14 to GRUAN data generally yields deviations close to 0 between parameterized and true $O_4$ VCDs for all stations, as shown in Fig. 8. Parameterized and true VCD show high correlations, indicating that the temporal variability of the atmospheric state is well captured by the simple parameterization based on surface values alone. The mean deviation $\delta_{RH}$ of all considered GRUAN profiles is $-0.3\%$, with a SD of $1.4\%$. For 11 out of the 17 stations, the mean agreement is within $1\%$. Largest deviations are found for La Reunion (REU), where $V_{O_4,\,RH}$ is biased low by $-2.5\%$. This is probably related to the altitude of this station of more than $2\,\mathrm{km}$ on a remote island in the Indian ocean. Highest positive deviation of $1.1\%$ is found for Barrow, with also highest SD of $1.8\%$. This is caused by some very high values during spring where surface temperatures are very low ($< 240\,\mathrm{K}$) and temperature inversions occur.

## 5 Discussion

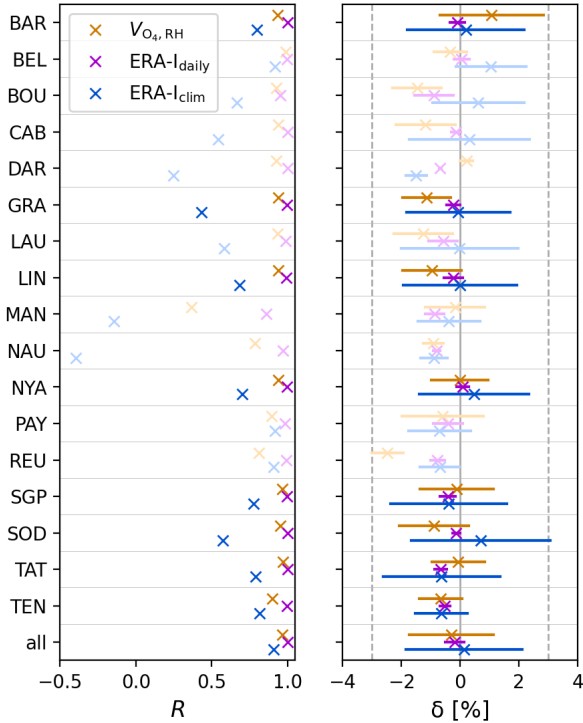

**Figure 8.** Comparison to the true $O_4$ VCD in terms of temporal correlation coefficient (left) and deviation $\delta$ according to Eq. 19 (right) for all GRUAN stations for $O_4$ VCDs calculated from (a) Eq. 14 based on GRUAN surface values (orange), (b) daily ERA-Interim profiles (purple) and (c) profiles from the ERA-Interim climatology (blue), interpolated to the GRUAN measurements in space and time. Light colors indicate stations with less than 100 profiles (compare Table B1).

## 5.1 Comparison to existing methods for the calculation of the $O_4$ VCD

Within MAPA (Beirle et al., 2019), the $O_4$ VCD was so far determined by integrating vertical profiles of the $O_4$ number density based on full profiles of $T$, $p$ and RH, which are by default taken from daily ERA-Interim simulations (ERA-I$_{daily}$), or, as fallback solution, from a monthly climatology complied from multi-annual ERA-Interim data (ERA-I$_{clim}$), both on 1° resolution.

Thus, we evaluate the performance of the proposed simple calculation of the $O_4$ VCD by comparing the results for GRUAN profiles, where the true $O_4$ VCD is known, also to the ERA-Interim profiles interpolated in space and time. In addition, a correction of surface altitude is necessary: For La Reunion, for instance, the radio sondes were launched at a surface altitude of 2 km, while the surface altitude in ERA-Interim (with 1° resolution) is just 54 m. This could easily cause deviations of 10% in $O_4$ VCDs when ignored. Thus we apply the following correction to the ERA-Interim profiles:

- In case of GRUAN station altitude being higher than ERA-Interim surface altitude, the ERA-Interim profiles of $T$ and RH are just linearly interpolated. As pressure profiles are almost exponential, $\ln(p)$ is linearly interpolated.

- In case of GRUAN station altitude being lower than ERA-Interim surface altitude, ERA-Interim profiles are extended by surface values of $T_0$ and $\ln(p_0)$ as derived from linear extrapolation of $T$ and $\ln(p)$, respectively. RH at ground, however, is not extrapolated, as this might result in unphysical values of RH below 0 or above 1. Instead, the value of the lowest ERA-Interim model layer is taken as $RH_0$.

We calculate the deviation from the true VCD (defined by the GRUAN profiles) according to Eq. 19 for daily and climatological ERA-Interim data. The correlation coefficients as well as mean and SD of the resulting deviations are also included in Fig. 8.

For $O_4$ VCDs based on daily ERA-Interim profiles, the agreement to VCDs integrated from GRUAN profiles is generally very good. Correlation coefficients are almost 1 and deviations are close to 0 for most stations. Only for mountainous sites as Boulder, where surface altitude differ between GRUAN and ERA-Interim, clear deviations from 0 are found.

Results based on the ERA-Interim climatology, however, show far weaker correlation than for daily ERA-Interim data, as they do not resolve day-to-day changes in meteorology. Mean deviations are within $\pm 1\,\%$ for most stations, with a SD of about $2\,\%$.

In comparison to these existing methods, the $O_4$ VCDs based on Eq. 14 are worse than those based on daily ERA-Interim profiles, but significantly better than those based on a profile climatology, in particular in terms of correlation and SD.

## 5.2 Impact of temperature inversions

The presented parameterizations derive the $O_4$ VCD just from surface values of $T$, $p$, and RH. This requires some basic assumptions about the atmospheric profile shape. In case of temperature inversions, these assumptions do not hold. Thus, we focused on daytime conditions by selecting only data with SZA<85°. But still, temperature inversions can also occur during daytime, in particular over cold water and ice surfaces, as well as shortly after sunrise.

Figure 9 displays temperature inversions, here defined as the difference between tropospheric maximum and surface temperature, for ERA5 data on 18 June 2018. Strong temperature inversions are found e.g. over Hudson Bay or the Great lakes where sea surface temperature is low. Also at the Western edge of the illuminated Earth (i.e. shortly after sunrise), temperature inversions occur, e.g. in North and South Africa at 6:00 UTC, indicating remainings of nocturnal profiles.

Large parts of the regions with high positive deviation $\delta_\Gamma$ (Fig. 3) or $\delta_{RH}$ (Fig. 7) actually correspond to temperature inversions. Thus, for MAX-DOAS measurements close to cold surface waters or other regions with temperature inversions, the formalism of Eq. 10 and Eq. 14 should only cautiously be applied, and corrections of surface temperature might be needed for better results.

As the impact of temperature inversions on $\delta_\Gamma$ is quite strong, we skip profiles with temperature inversions of more than 2 K for the investigation of the effects of humidity (Sect. 5.3) and surface altitude (Sect. 5.4) in order to avoid interference of different effects.

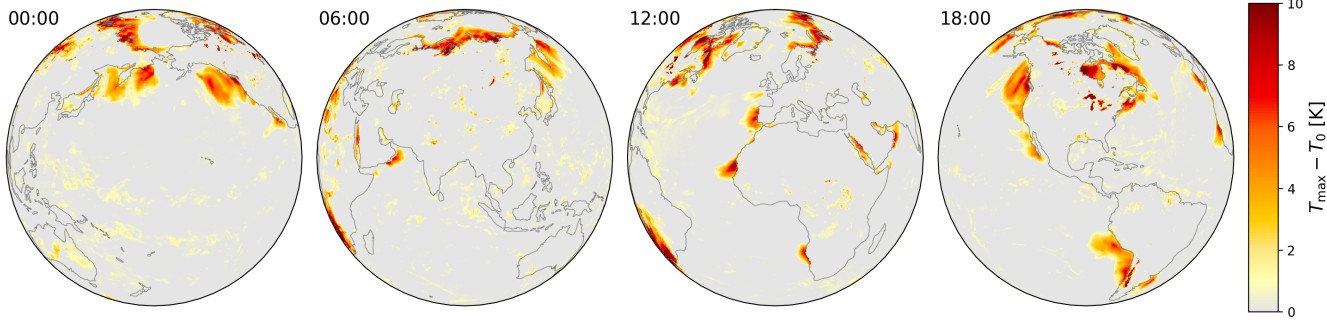

**Figure 9.** Temperature inversions, expressed as difference between tropospheric maximum and surface temperature, for ERA5 simulations at 0:00, 6:00, 12:00, and 18:00 UTC on 18 June 2018.

## 5.3 Impact of humidity

The formalism in Sect. 2.3 is assuming dry air. Addition of humidity results in lower $O_2$ and $O_4$ number densities, which
significantly affects the $O_4$ VCD, especially in the tropics (Wagner et al., 2019). Humidity affects all terms in Eq. 6 (i.e. the $O_2$ VCD, the $O_2$ surface number density, and the ratio of effective heights of $O_2$ and $O_4$), but cannot be accounted for in the formalism without completely losing the simplicity of Eq. 10.

However, these effects are partly accounted for in Eq. 14, with empirically determined parameters $a$ and $b$, since the ratio $Q$ was determined based on the true $O_4$ VCD where humidity effects were appropriately accounted for.

In order to check for possible remaining impacts of humidity on the performance of Eq. 14, we check how far $\delta_{RH}$ is related to specific humidity at ground (Fig. 10). In addition, we compare $\delta_{RH}$ also to the total column water vapor, as this provides information on humidity in the full column, not only at surface. In both cases, correlations are low, and no significant impact of humidity on $\delta_{RH}$ could be found.

## 5.4 Impact of surface altitude

Figures 6 and 7 reveal systematic spatial patterns in $\delta_{RH}$ corresponding to mountains. We thus investigate a possible relation between surface altitude and $\delta_{RH}$ for all investigated datasets (Fig. 11).

For the WRF simulations, the Alps can be clearly recognized in Fig. 6, with mountains showing lower values of $\delta_{RH}$. This can also be seen in the density plot in Fig. 11 (a), where surface altitude and $\delta_{RH}$ are anticorrelated with R$= -0.46$, and a decrease of $\delta_{RH}$ of roughly $1\,\%$ per km. For GRUAN stations (c), results are similar, but statistics are poor, and the correlation
coefficient is low, as only two stations (Boulder and La Reunion) are available with a surface altitude above $1\,\text{km}$.

For ERA5, however, results are not as clear as those for WRF. The correlation coefficient is low, and for altitudes between 2 and 3 km, it looks like $\delta_{RH}$ is increasing rather than decreasing with altitude. And for very high surface altitudes as found over the Himalaya, $\delta_{RH}$ is still close to 0 and would not match the slope of $1\,\%$ per km estimated for WRF.

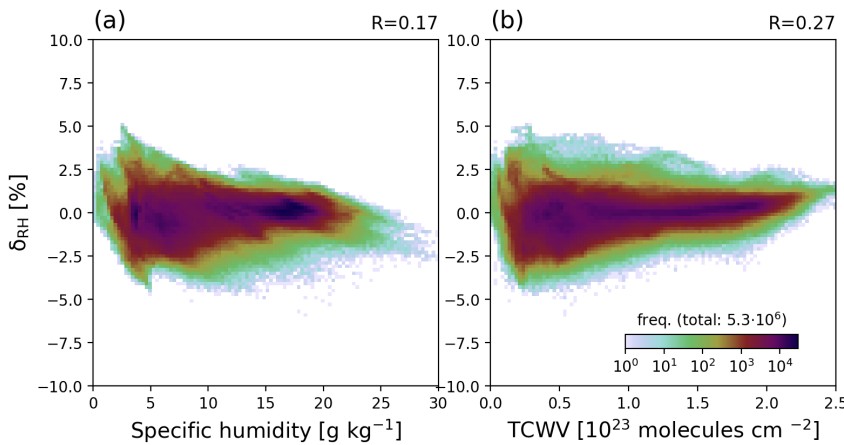

**Figure 10.** Deviation $\delta_{RH}$ according to Eq. 19 as function of (a) specific humidity at ground, and (b) TCWV for ERA5 simulations on 18 June 2018. Temperature inversions with $T_{max} - T_0 > 2\,\text{K}$ have been skipped.

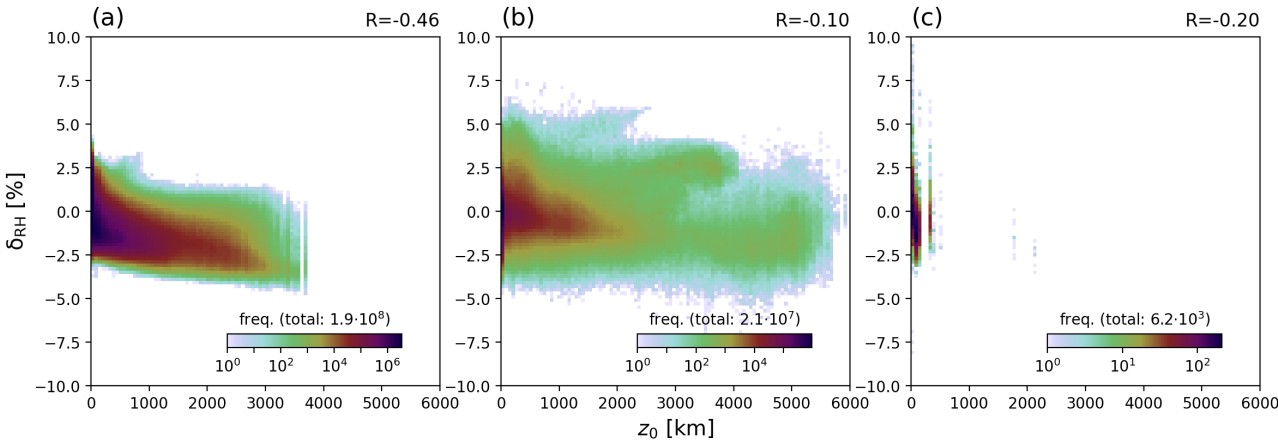

**Figure 11.** Dependency of $\delta_{RH}$ on surface altitude for (a) WRF data for May to June 2018, (b) ERA5 data for all selected days, and (c) GRUAN data. For (b), Temperature inversions with $T_{max} - T_0 > 2\,\text{K}$ have been skipped.

The reason for the poor correlation between $z_0$ and $\delta_{RH}$ for ERA5 compared to the WRF results is not clear to us. Obviously, other factors would probably also have to be considered (season, SZA). But since there is no clear correlation, and a quantitative correction would rather worsen $\delta_{RH}$ instead of improving it for several mountain areas around the globe, we decided not to apply an explicit correction for surface altitude.

Consequently, the parameterization of Eq. 14 has higher uncertainties when applied for mountainous sites: for $z_0 > 2\,\text{km}$, $\delta_{RH}$ is $-0.5\%$ on average with a SD of $1.8\%$. But still, the parameterized $O_4$ VCD matches the requirement of accuracy/precision better than $3\%$ even for elevated sites.

## 5.5 Diurnal cycles

Surface conditions can change rapidly, e.g. in case of passing frontal systems or storm tracks. For such rapid changes, the change of the true $O_4$ VCD might not be adequately represented by the change of $V_{O_4, RH}$. These effects are reflected in the SD of deviations $\delta_{RH}$ for ERA5, WRF, and GRUAN.

In addition, surface values could change *systematically* during the day in case of strong solar irradiation, causing a diurnal cycle of surface temperature and the $O_4$ VCD (Wagner et al., 2019). Thus we investigate the diurnal cycles of $T_0$, $p_0$, $RH_0$, and the respective $O_4$ VCDs $V_{O_4, RH}$ and $V_{O_4, true}$ in more detail, and check how far (a) the WRF simulations reflect the actual diurnal cycles and (b) the parameterized $O_4$ VCDs based on surface values reflect the diurnal cycle of the true $O_4$ VCD. For this, we extract the WRF simulations at the locations of the DWD ground station network. In order to focus on strong diurnal

patterns, we select at each station those days where the intra-day change of surface temperature, as recorded by DWD, exceeds 10 K.

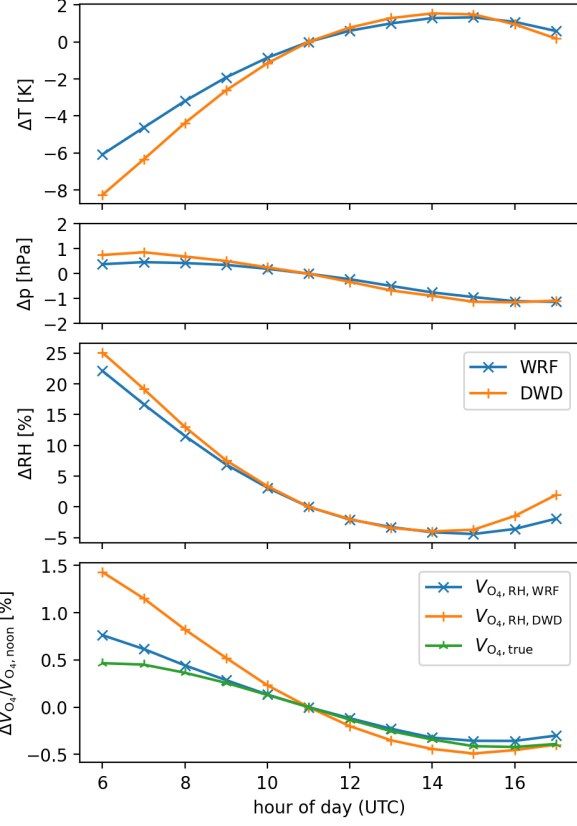

**Figure 12.** Diurnal cycles of surface temperature (a), pressure (b), RH (c), and the $O_4$ VCD (d). Data points show the mean values for all stations for May to June 2018 for days where the increase in $T_0$ (from DWD) over the day is larger than 10 K. For better comparison, the mean value at 11:00 UTC (around solar noon for Germany) is subtracted from all datasets. For the $O_4$ VCD, the relative change is shown.

Figure 12 displays the diurnal cycles of surface properties and $O_4$ VCDs for WRF and DWD station data. While surface pressure shows no relevant changes during day, surface temperature increases by $9.8\,\mathrm{K}$ from morning to afternoon due to the selection of days with strong diurnal cycle in surface temperature[2]. For WRF simulations, a similar pattern is found, but the mean temperature increase over the day is smaller ($7.4\,\mathrm{K}$). As $V_{O_4,\,RH}$ is reciprocal to $T_0$, a change of $10\,\mathrm{K}$ in surface temperature alone would correspond to a change of $V_{O_4,\,RH}$ of $3.5\,\%$. However, at the same time, RH decreases by about $30\,\%$, which has an opposite effect on $V_{O_4,\,RH}$. Consequently, the diurnal cycle of $V_{O_4,\,RH}$ is only moderate (about $1.9\,\%$ and $1.1\,\%$ decrease from morning to evening for DWD and WRF, respectively, where the cycle for WRF is less strong due to the less strong cycle in $T_0$).

The true $O_4$ VCD, as derived from the integrated WRF profiles, also decreases over the day, and agrees well to $V_{O_4,\,RH}$ (WRF) in the afternoon. In the morning, however, $V_{O_4,\,RH}$ (WRF) is higher compared to noon by $0.8\,\%$, while $V_{O_4,\,true}$ is only $0.5\,\%$ higher. This deviation between parameterized and true $O_4$ VCD indicates that in the early morning, surface measurements are not as useful for determining the full column, which is probably related to remainders of the nocturnal boundary layer which often has atypical lapse rates due to temperature inversions.

But even during morning hours, the systematic error made by $V_{O_4,\,RH}$ is relatively small, at least for the investigated time period for Germany. But also for the global ERA5 analysis, the impact of diurnal cycles on the calculation of the $O_4$ VCD is only moderate; otherwise, Figures 7 and C2 would show systematic East-West gradients.

Thus, the parameterization of Eq. 14 also reflects most of the diurnal cycle of the $O_4$ VCD, with remaining systematic errors below $0.3\,\%$.

## 5.6 Accuracy and precision

In Eq. 14, we provide a formula for the calculation of the $O_4$ VCD. Accuracy and precision of the resulting $V_{O_4}$ thereby depend on accuracy and precision of (1) the chosen parameterization and (2) surface values $p_0$, $T_0$ and $RH_0$.

1. We estimate overall accuracy and precision of Eq. 14 to $< 1\,\%$ and $< 2\,\%$ based on mean and SD of deviations between parameterized and true $O_4$ VCD for WRF, ERA5 and GRUAN data as presented above. Higher deviations can occur in particular in case of temperature inversions (see Sect. 5.2).

2. Application of Eq. 14 requires surface measurements of $p_0$, $T_0$, and $RH_0$. Uncertainties of temperature and pressure are rather uncritical, as an error of $1\,\mathrm{K}$ and $1\,\mathrm{hPa}$ for $T_0$ and $p_0$ would correspond to an error of $0.3\,\%$ and $0.2\,\%$ in $V_{O_4,\,RH}$, respectively. In order to reach an accuracy/precision of $1\,\%$, the corresponding errors of $RH_0$ have to be lower than $16\,\%$. These limits should be achievable for adequate meteorological instrumentation and a measurement procedure following WMO guidelines. In particular, surface temperature should be measured at about $1.25$ to $2\,\mathrm{m}$ above ground using a radiation shield (WMO, 2018).

---

[2]Note that the mean change is lower than the threshold used for the selection of DWD stations. This is caused by averaging diurnal cycles with maxima occuring at different times of the day.

The proposed parameterization thus allows to calculate $O_4$ VCD with overall uncertainties below $3\%$, which is sufficient for applications in MAX-DOAS profile inversions. Compared to existing methods, the parameterization yields even better results than a profile climatology.

We thus consider the proposed parameterization as useful approach for determining the $O_4$ VCD for cases where no daily model profiles are available, and recommend to also apply it for mountain sites for comparison and possible correction of daily model profiles.

## 6    Conclusions

The $O_4$ VCD can be expressed in terms of surface pressure and temperature based on physical laws, if a constant lapse rate is
assumed, without the need for constructing full vertical profiles. With an empirical correction which parameterizes the effective lapse rate as linear function of surface RH, we could present a formula for simple and quick calculation of the $O_4$ VCD based on $p_0$, $T_0$, and $RH_0$:

$$V_{O_4,RH} = \frac{6.733 \cdot 10^{39}}{1.774 + 0.1182 \cdot RH_0} \cdot \frac{p_0^2}{T_0} \ \mathrm{molec}^2 \ \mathrm{cm}^{-5}. \tag{20}$$

This parameterization reproduces the real $O_4$ VCD, as derived from vertically integrated profiles, within $-0.7\% \pm 1.2\%$ for
WRF simulations around Germany, $0.2\% \pm 1.8\%$ for global reanalysis data (ERA5), and $-0.3\% \pm 1.4\%$ for radiosonde soundings around the world. Largest deviations are observed in case of temperature inversions which cause too low $T_0$ (compared to the remaining profile) and thus high biased estimates of $V_{O_4,\,RH}$. For applications to measured surface values, uncertainties of $1\,\mathrm{K}$, $1\,\mathrm{hPa}$, and $16\%$ for temperature, pressure, and RH correspond to relative uncertainties of the $O_4$ VCD of $0.3\%$, $0.2\%$, and $1\%$, respectively.

This accuracy and precision of $< 3\%$ is typically lower than other uncertainties of spectral analysis or radiative transfer modeling (Wagner et al., 2019). Thus, the proposed parameterization is well suited for application in MAX-DOAS profile inversions. Moreover, the parameterization reflects the true $O_4$ VCD, as derived from radiosonde measurements, even better (in particular in terms of temporal correlation and SD) than $O_4$ VCD calculated from a climatology of atmospheric profiles of $T$, $p$ and RH. We thus recommend to equip MAX-DOAS measurement stations with state-of-the-art thermometer (with
radiation shield), barometer, and hygrometer.

*Code availability.* A Python implementation of the derived functions for the calculation of the $O_4$ VCD is provided in the Supplementary material.

## Appendix A: Ratio of effective heights

The ratio of the effective heights for $O_4$ and $O_2$ in Eq. 6 depends on the shape of the $O_2$ profile. For specific shapes the ratio can be calculated explicitly. Below, we derive the ratio $\frac{h_{O_2}}{h_{O_4}}$, which allows for simpler notation avoiding compound fractions. For application in Eq. 6, the inverse ratio has to be taken.

### A1 Isothermal atmosphere

For the simple assumption of a barometric pressure profile with constant $T$, the $O_2$ number density decreases exponentially with altitude:

$$n_{O_2} = n_{O_2,0} \cdot \exp\left(-z'/H\right) \tag{A1}$$

with the scale height $H$. In this case, the integral of Eq. 3 directly yields $H$, i.e. the effective height equals the scale height for exponential profiles. For $O_4$, the profile is exponentially decreasing as well, with the scale height being half of that for $O_2$. Thus, for $O_2$ profiles declining exponentially with $z$, the ratio of effective heights is just

$$\frac{h_{O_2}}{h_{O_4}} = 2. \tag{A2}$$

### A2 Polytropic atmosphere

If the temperature is changing linearly with altitude, i.e. the dependence of $T(z) = T_0 + \Gamma \cdot (z - z_0)$ is described by a constant lapse rate $\Gamma$, the resulting profile of $O_2$ follows a power function:

$$n_{O_2} = n_{O_2,0} \cdot \left(1 + \frac{\Gamma}{T_0} z'\right)^{-\alpha}, \tag{A3}$$

with

$$z' = z - z_0 \tag{A4}$$

being altitude above surface, and

$$\alpha = 1 + \frac{g \cdot M}{R \cdot \Gamma} \tag{A5}$$

being the constant exponent.

Note that for a constant lapse rate, temperature reaches 0 K at an altitude of

$$z_{\text{TOA}} = \frac{T_0}{\Gamma} \tag{A6}$$

For $T_0 = 300\,\text{K}$ and $\Gamma = -6.5\,\text{K km}^{-1}$, $z_{\text{TOA}}$ is about 46 km.

Thus, Eq. A3 is defined from $z' = 0$ to $z' = z_{\text{TOA}}$, and $n_{O_2}$ is set to 0 above.

Integration of Eq. 3 yields

$$h_{O_2} = \int_0^{z_{TOA}} \left(1 + \frac{\Gamma}{T_0} z'\right)^{-\alpha} dz'$$

$$= \left[\frac{1}{-\alpha+1} \left(1 + \frac{\Gamma}{T_0} z'\right)^{-\alpha+1} \cdot \frac{T_0}{\Gamma}\right]_0^{z_{TOA}}$$

$$= \frac{1}{-\alpha+1} \cdot \frac{T_0}{\Gamma} \tag{A7}$$

For $O_4$, the number density profile is

$$n_{O_4} = n_{O_4,0} \cdot \left(1 + \frac{\Gamma}{T_0} z'\right)^{-2\alpha}, \tag{A8}$$

and thus

$$h_{O_4} = \frac{1}{-2\alpha+1} \cdot \frac{T_0}{\Gamma}. \tag{A9}$$

The ratio of effective heights can then be calculated as

$$\frac{h_{O_2}}{h_{O_4}} = \frac{2\alpha - 1}{\alpha - 1}$$

$$= \frac{2\frac{g \cdot M}{R \cdot \Gamma} + 1}{\frac{g \cdot M}{R \cdot \Gamma}}$$

$$= 2 + \frac{R}{g \cdot M} \cdot \Gamma. \tag{A10}$$

For a lapse rate of 0 this equals the result for the isothermal atmosphere ($\equiv 2$). For a typical lapse rate of $-6.5\,\mathrm{K\,km^{-1}}$, the ratio of effective heights is 1.81.

## A3 Impact of the tropopause

In the previous section, the ratio of effective heights was calculated assuming a constant lapse rate throughout the atmosphere. A more realistic approach would be to assume a constant temperature above the tropopause (TP), as was done in Wagner et al. (2019). However, with the separation of the atmosphere in troposphere and stratosphere, it would not be possible to express the ratio of effective heights as simple function of the lapse rate as in Eq. A10. Thus, we decided to neglect the impact of the tropopause on the ratio of effective heights in the derivation of Eq. 10.

This causes a bias of $V_{O_4,\Gamma}$ that can be easily quantified from Eq. 10 itself (applied at the tropopause instead of ground): The stratospheric $O_4$ column for constant $T$ is $\frac{C}{2} \cdot \frac{p_{TP}^2}{T_{TP}}$, while it is $\frac{C}{2 + \frac{R}{g \cdot M}\Gamma} \cdot \frac{p_{TP}^2}{T_{TP}}$ for constant lapse rate. The difference is $6 \cdot 10^{40}$ molecules$^2$ cm$^{-5}$ (for $T_{TP} = 200\,\mathrm{K}$, $p_{TP} = 193\,\mathrm{hPa}$), which is about 0.45% of the total $O_4$ VCD.

Thus, the $O_4$ VCD derived from Eq. 10 is higher than the respective VCD resulting from the profile construction proposed in Wagner et al. (2019). For $V_{O_4,RH}$ (Eq. 14), this bias is eliminated by the empirical fit of the parameters $a$ and $b$ to the true $O_4$ VCD.

## A4  Side note: Determining the effective lapse rate from direct sun measurements of $O_2$ and $O_4$

The $O_4$ VCD depends on the ratio of effective heights for $O_2$ and $O_4$ (Eq. 6), which can be expressed by the atmospheric lapse rate (Eq. A10). This formalism might also be used in the other direction: from total column measurements of $O_2$ and $O_4$, an effective atmospheric lapse rate can be derived.

$$2 + \frac{R}{g \cdot M} \cdot \Gamma \overset{(A10)}{=} \frac{h_{O_2}}{h_{O_4}} \overset{(6)}{=} \frac{V_{O_2}}{V_{O_4}} \cdot n_{O_2,0} \overset{(8)}{=} \frac{V_{O_2}}{V_{O_4}} \cdot \frac{\nu_{O_2} \cdot p_0}{R \cdot T_0} \tag{A11}$$

and thus

$$\Gamma = \left( \frac{V_{O_2}}{V_{O_4}} \cdot \frac{\nu_{O_2} \cdot p_0}{R \cdot T_0} - 2 \right) \cdot \frac{g \cdot M}{R} \tag{A12}$$

This formalism might be applied to direct sun measurements, where light paths are well defined by the SZA. Even for limited accuracy of column measurements of $O_2$ and $O_4$, this would allow to derive time series of an effective lapse rate, reflecting the state of the lower atmosphere.

# Appendix B: Datasets

## B1  Regional model and surface measurements

### B1.1  WRF simulations

A nested domain centred at 49.12° N, 10.20° E was set up in Lambert conformal conic (LCC) projection with coarser domain (d01) at $15 \times 15\ \mathrm{km}^2$ horizontal resolution and finer domain (d02) at $3 \times 3\ \mathrm{km}^2$ resolution (Fig. B1). The spatial extent of the d01 domain is $4800 \times 3416\ \mathrm{km}^2$ while that for d02 is $1578 \times 1473\ \mathrm{km}^2$. Vertically, the model extends from surface until $50\ \mathrm{hPa}$ with 42 terrain following layers in between. For constraining the meteorological initial and lateral boundary conditions, we use the ERA5 reanalysis dataset with a horizontal resolution of 0.25°×0.25° and a temporal resolution of 3 hours, downloaded at pressure levels and at the surface. The soil classification, terrain height, and land use patterns were taken from the 21 category Noah-modified IGBP-MODIS land use data.

The model simulations were set up for May and June in 2018. The selection of data with SZA$< 85°$ results in a daily coverage from $6{:}00\ \mathrm{h}$ to $17{:}00\ \mathrm{h}$ UTC. Here we focus on model profiles in the d02 domain. The partial column of $O_4$ above 50 hPa is considered accordingly in the calculation of the true $O_4$ VCD (see Sect. 2.4).

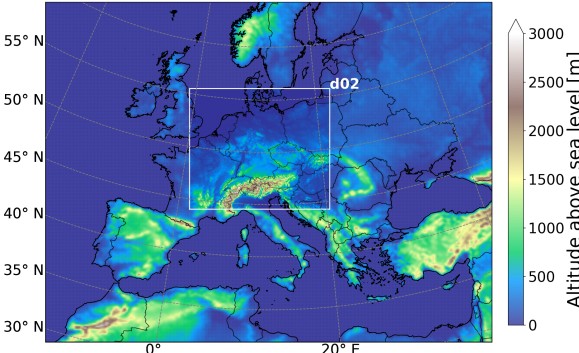

**Figure B1.** Nested model domains d01 (full figure) and d02 (marked pane) of the WRF simulations.

### B1.2  DWD weather stations

Germany's National Meteorological Service (Deutscher Wetterdienst, DWD) provides hourly measurements of surface temperature, pressure and relative humidity for a network of ground stations in Germany (Kaspar et al., 2013). Data are provided via the climate data center web interface (CDC-v2.1; https://cdc.dwd.de/portal/). The meteorological measurements are performed in accordance to the guidelines of the world meteorological organization (WMO) to minimize local effects. Additionally, we have applied quality control filters such that the QUALITAETS_BYTE (QB) is below 4 (thereby excluding untested, objected, and calculated values), and QUALITAETS_NIVEAU (QN) is either 3 (automatic control and correction) or 7 (second control done, before correction) to only retain measurements of high quality. By applying these criteria, we retained 98.2 %, 100%,

and 99.5% of $T_0$, $p_0$, and $RH_0$ data, respectively. Note that using only data with QN=10 (the best possible quality check level) would result in no data left for the period considered in this study. If only QN=7 had been applied, we would have retained the same number of $T_0$ and $RH_0$ but no $p_0$ data.

For this study, we extract DWD measurements for May to June 2018, 6:00 to 17:00 UTC, and only consider stations provid-
ing $T_0$, $p_0$, and $RH_0$ simultaneously, resulting in 206 stations which are displayed in Fig. B2.

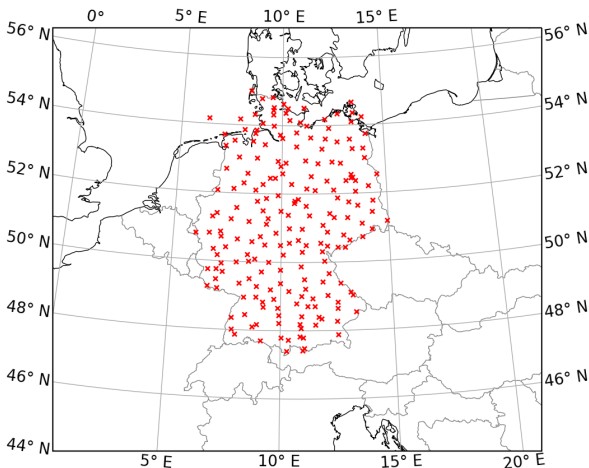

**Figure B2.** Location of the 206 DWD ground stations providing simultaneous measurements of surface values of $T$, $p$ and RH during May to June 2018.

### B1.3   Validation of WRF surface values

We use the DWD network of surface stations for investigating the accuracy and precision of the WRF simulations. Figure B3 displays correlations between surface values from the DWD station network and the respective WRF simulations. For this purpose, each station is associated with the nearest neighbor from the WRF simulation. We do not interpolate the WRF data as
we still want to compare the parameterized $O_4$ VCD with the true VCD derived from vertical integration of the WRF profiles.

Surface altitude (a) is lower in the gridded elevation map used as input in the WRF simulations by $20\,\mathrm{m}$ on average, and by almost $1\,\mathrm{km}$ for the station on Germany's highest mountain, Zugspitze. This is a consequence of the spatial resolution of the WRF simulations of $1\,\mathrm{km}$, which is not sufficient for resolving single mountains. The systematic negative bias of WRF surface altitude indicates that the DWD stations tend to be located on hill and mountain tops.
This difference in altitude would directly affect the comparisons of $T$ and particularly $p$. Thus, we apply a simple correction of station values and extrapolate them to the respective WRF surface altitude assuming a lapse rate of $-6.5\,\mathrm{K}\,\mathrm{km}^{-1}$. For RH, no correction is applied.

The comparison reveals a good agreement between surface values from WRF and DWD, with remaining systematic biases of WRF simulations of $-1\,\mathrm{K}$ for $T_0$ and $1\,\%$ for $RH_0$.

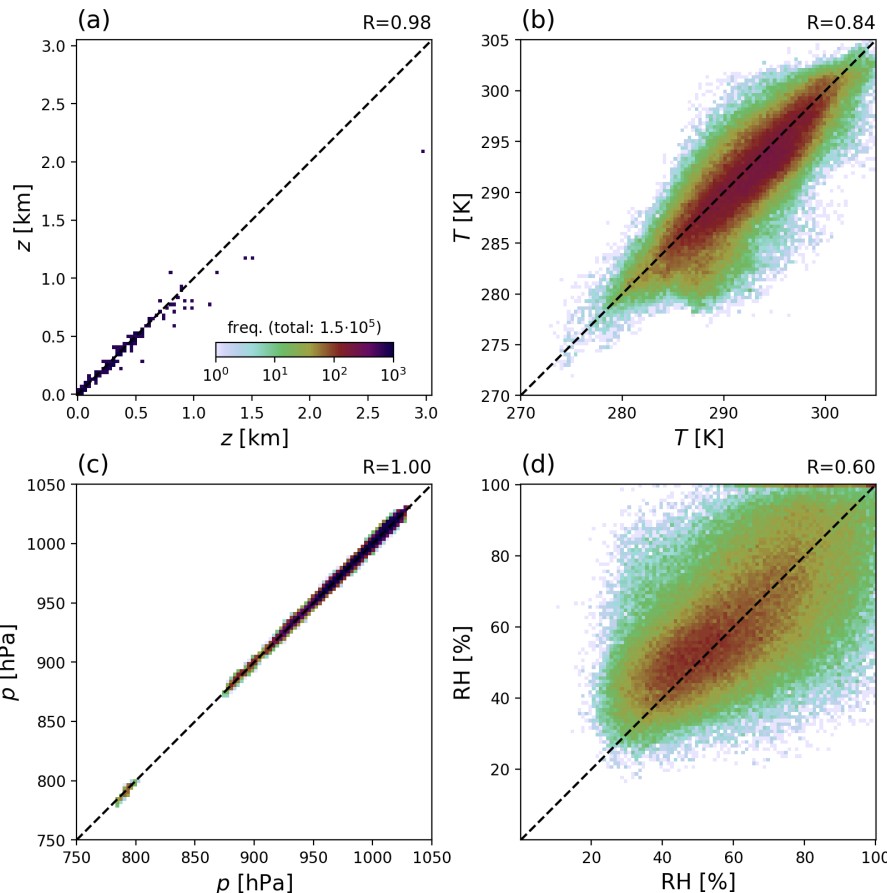

**Figure B3.** Comparison of WRF surface values ($y$ axis) to DWD ground stations ($x$ axis). For $T$ and $p$, station values are adjusted to the mean altitude of the respective gridded elevation map used as input for WRF siumulations (see text for details).

## B2    GRUAN stations

The GRUAN stations used in this study are listed in Table B1, including station shortcut and full name, latitude, longitude, altitude of the station, and the number of available profiles with SZA< 85°. Figure B4 displays a map showing the GRUAN station locations.

The temporal cover of radio sonde measurements at the different stations is displayed in Fig. B5. Note that some stations only contribute a low number of measurements. Still, we decided to keep all stations, as the application of a threshold for a minimum number of profiles of e.g. 50 would remove all tropical sites (Darwin, Manus and Nauru).

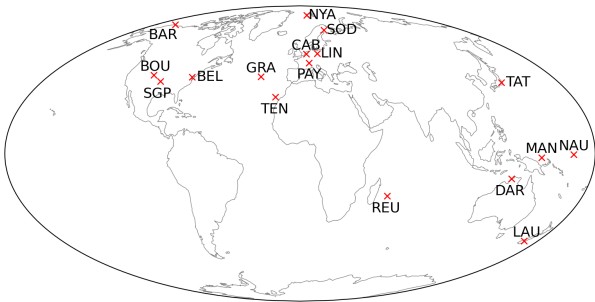

**Figure B4.** Location of GRUAN stations considered in this study. For station names and further details see Table B1.

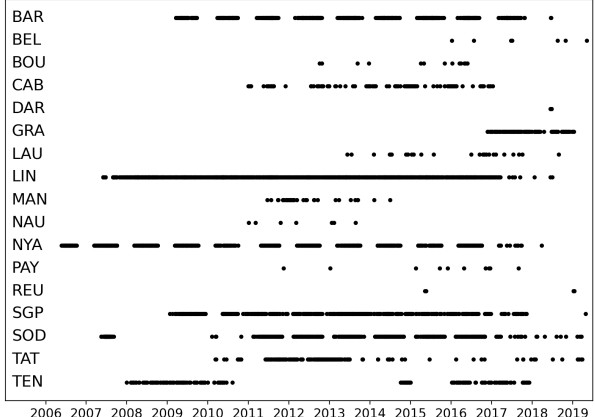

**Figure B5.** Time of the available sonde flights (with SZA<85°) for the GRUAN stations considered in this study. For station names and further details see Table B1.

**Table B1.** List of GRUAN stations and number of available sonde flights (only considering SZA<85°) used in this study.

| Label | Name | Lat [° N] | Lon [° E] | $z_0$ [m] | Profiles |
|-------|------|-----------|-----------|-----------|----------|
| BAR | Barrow | 71.32 | -156.62 | 8 | 1189 |
| BEL | Beltsville | 39.05 | -76.88 | 53 | 7 |
| BOU | Boulder | 39.95 | -105.20 | 1743 | 13 |
| CAB | Cabauw | 52.10 | 5.18 | 1 | 98 |
| DAR | Darwin | -12.42 | 130.89 | 35 | 4 |
| GRA | Graciosa | 39.09 | -28.03 | 30 | 125 |
| LAU | Lauder | -45.05 | 169.68 | 371 | 25 |
| LIN | Lindenberg | 52.21 | 14.12 | 103 | 2255 |
| MAN | Manus | -2.06 | 147.43 | 4 | 42 |
| NAU | Nauru | -0.52 | 166.92 | 7 | 7 |
| NYA | NyAlesund | 78.92 | 11.92 | 15 | 1059 |
| PAY | Payerne | 46.81 | 6.95 | 491 | 10 |
| REU | LaReunion | -21.08 | 55.38 | 2156 | 8 |
| SGP | Lamont | 36.61 | -97.49 | 315 | 566 |
| SOD | Sodankyla | 67.37 | 26.63 | 179 | 602 |
| TAT | Tateno | 36.06 | 140.13 | 30 | 165 |
| TEN | Tenerife | 28.32 | -16.38 | 121 | 163 |

## Appendix C: Additional ERA5 results

Figures C1 and C2 display additional results for $\delta_\Gamma$ and $\delta_{RH}$, respectively, for 18 March and 18 September 2018.

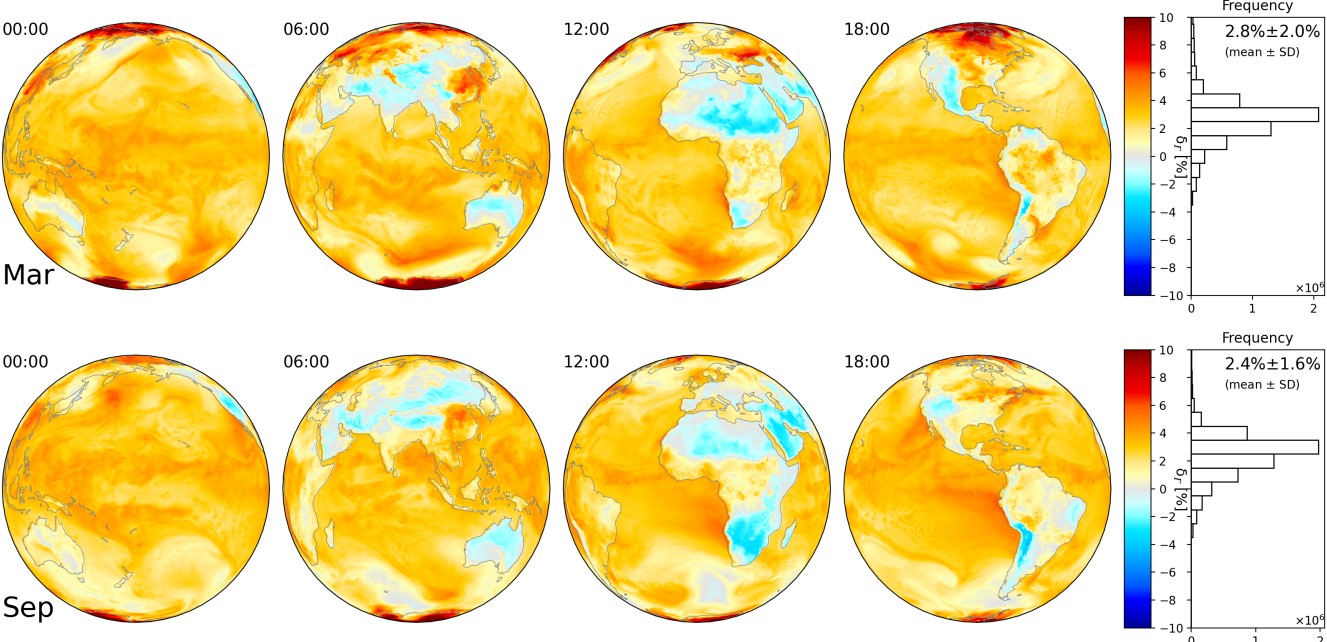

**Figure C1.** Deviation $\delta_\Gamma$ according to Eq. 19 for ERA5 at 0:00, 6:00, 12:00 and 18:00 UTC on 18 March (top) and 18 September (bottom) 2018. The projection focuses on daytime for each timestep. On the right, the frequency distribution and mean and SD are given for all hourly outputs of the respective day.

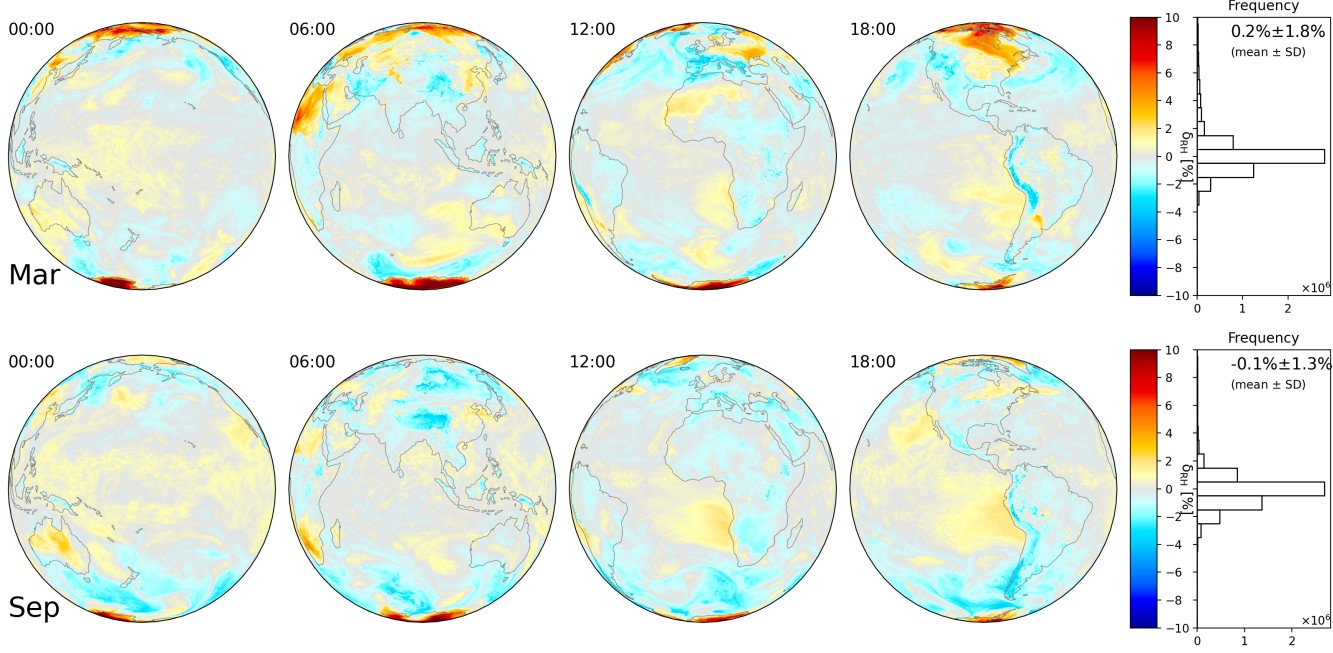

**Figure C2.** Deviation $\delta_{\mathrm{RH}}$ according to Eq. 19 for ERA5 at 0:00, 6:00, 12:00 and 18:00 UTC on 18 March (top) and 18 September (bottom) 2018. The projection focuses on daytime for each timestep. On the right, the frequency distribution and mean and SD are given for all hourly outputs of the respective day.

*Author contributions.* CB initiated this study by proposing to express the $O_4$ VCD by surface number density and column density of $O_2$. VK performed the WRF simulations and preprocessed the DWD data. SD processed ECMWF data. SD, CB and TW provided input on $O_4$ VCD calculation and meteorology. SB developed the full formalism, performed the intercomparisons to external datasets, and wrote the manuscript, with input and feedback from all co-authors.

*Competing interests.* None.

*Acknowledgements.* We would like to thank Rajesh Kumar (UCAR Boulder) and Sergey Osipov and Andrea Pozzer (both MPIC Mainz) for support in setting up the WRF simulations. We thank MPCDF Garching for providing computation resources for the WRF simulations. ERA Interim and ERA5 data used in this study are provided by the European Center of Medium-Range Weather Forecasts (ECMWF). Meteorological data for ground stations in Germany were provided by Deutscher Wetterdienst (German Weather Service, DWD). Radiosonde measurements were provided by the GRUAN network.

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
