# Peer review of "Calculating the vertical column density of $O_4$ during daytime from surface values of pressure, temperature and relative humidity"

_Atmospheric Measurement Techniques, 2021_

## Author Comment (AC1)

**Reply to reviewer #1**

We provide a point-by-point reply to the issues raised by reviewer 1 below. The original review is included in grey. Text changes in the manuscript are indicated in italic font.

Please also note the modifications made in the revised manuscript that are specified in the Corrigendum.

1. General comments
The manuscript "Calculating the vertical column density of $O_4$ from surface values of pressure, temperature and relative humidity" presents in varying level of detail the derivation of, and the validation of, a daytime applicable method to calculate the $O_4$ vertical column density from surface values of pressure, temperature and relative humidity. Hence, the title is well chosen. The main application area of this approximation are parametrized profile inversion algorithms for MAX-DOAS measurements. The authors also mention possible benefits for optimal estimation based profile inversion algorithms, stemming from the high correlation of surface relative humidity and temperature effective lapse rate. I recommend this manuscript for publication after intermediate revisions.

We thank the reviewer for the elaborated and thorough feedback to our study. We carefully considered the issues raised by the reviewer, and in many cases, they helped us to improve the paper. In some cases, however, we have a different point of view.
Below, we deal with the reviewer comments point by point. In cases where we disagree with the reviewer's evaluation, we motivate our point of view in this reply and in the revised manuscript.

While there is certainly scientific value in the presented method, the quality of the presentation and the structure of the manuscript, have to be improved. The validation of the method is not quite sufficient and has to be extended. The degree of explicit derivation of equations varies from "very detailed" to "almost insufficient" and should be brought to a more "equal" level.
We will refer to these aspects below when they are concretized by the reviewer.

The presented derivation of Eq. 9 seems unnecessarily complicated: Starting at the usual barometric formula for the air density for an atmosphere with constant laps rate
$$\rho = \rho_0 \ (T_0/(T_0 + \Gamma(x-x_0)))^{1+(gM/R/\Gamma)}] = \rho_0 \ (1+ \Gamma x'/T_0)^{-(gM/R/\Gamma)}$$
($x'=x-x_0$) and integrating the square of this $\rho$ multiplied by the oxygen volume mixing ratio (i.e. integrate the $O_4$ density) from the surface (0) to infinity, replacing $\rho_0$ by $p_0/T_0/R$, directly yields Eq. 9. Maybe the authors can comment on why they chose to over complicate things with introducing the ratio of $h_{O2}$ and $h_{O4}$. (Likewise, choose the density formula for 0 lapserate and integrate the square, in order to arrive at the corresponding equation for 0 lapse rate). I highly recommend to streamline this. I cannot see any added benefit of the method used by the authors, but I do see a lot of unnecessary turns given.

Equation 9 could indeed be derived directly, if a constant lapse rate is assumed. However, the concept of a constant lapse rate is a significant simplification, and real atmospheric profiles are more complex.
As pointed out in section 2.2, Eq. 6 is derived without any simplification or additional assumptions. Thus, the formalism derived in Equations 1 to 6 holds for any atmosphere, as we now state explicitly in the revised manuscript.
Equation 9 is a special case of Eq. 6 for constant lapse rate. For real atmospheric profiles, where lapse rate is usually not constant, still the formalism of Eq. 9 can be applied for the *effective* lapse rate, which was defined in Appendix A. This definition can only be understood in the context of equations 1 to 6. In the revised manuscript, we clarify this aspect by providing the definition of the effective lapse rate already in subsection 2.3.

At several occasions in the manuscript, the authors refer to later sections or to, at that point,

unproven and not referenced statements. This makes it impossible to read the manuscript in a linear fashion.The main line of reasoning should be clearly stated and followed.

(*) We agree that reference to "future" sections is suboptimal, but we consider it sometimes unavoidable, as the line of arguments does not always follow a linear fashion. In addition, we think that it is not unusual to e.g. refer already in the result section to a specific aspect that will be discussed in more detail in the discussion section.

We decided to organize the manuscript having a section on formalism, followed by data sets and applications. Thus it is unavoidable that sometimes the motivation for choices made in the formalism is not directly supported by data. However, the alternative would be to jump forth and back several times between formalism, results, and discussions, which we do not consider as a better alternative.

In the revised manuscript, we have slightly revised the order of subsections in a more plausible order, and tried to minimize references to the "future" as far as possible.

Several statements are made without proof or proper reference.

We now support the respective statements with additional figures, concrete numbers or references.

Regarding style, the guidelines of AMT are, in several aspects, not followed.

We have adopted AMT guidelines in the revised manuscript.

The quality of the plots is mostly ok but should also be improved before final acceptance (especially the readability of axis labels).

We have revised the figures and increased the font size of axis labels.

Apart from the unfortunate structuring of the manuscript and the unnecessary turns given in order to arrive at the important equation, the biggest point of criticism is perhaps on the method validation and the lack of showing the improvement when using this new method over other methods to estimate the $O_4$ VCD, as well as the actual effect on the final product, the retrieved AOD.

Three rather limited data sets were used for validation. Each of these datasets needs to be extended.

We are surprised by the evaluation of the reviewer. In the AMTD study, we have applied the derived formalism to ~1e7 profiles from ECMWF, 3e7 from WRF simulations, and 6000 GRUAN sonde profiles. In the revised manuscript, we doubled the number of considered days from ECMWF by adding one day from autumn and spring, and we extended the application to WRF data to the full 2-month simulation period, increasing the number of WRF profiles to almost 2e8.

Variability in space as well as fluctuations due to "weather" (high/low pressure systems) is well covered by the global ECMWF simulations. Variability in time is covered for several GRUAN stations. The derived statistical quantities are robust: standard errors of mean and SD are close to zero. From the different datasets, we derived quite consistent numbers for mean and SD. Thus, we consider the presented data to be sufficient for estimating realistic numbers for the errors made by the parameterisation.

For one of the datasets (global model), half of the dataset was used to fit parameters in the model; still, that same half was also used for validation. This should really be avoided. It is advisable to add a separate day for parameter fitting.

The calculation of $\delta_{RH}$ for ECMWF data from 18 June is using the same data as used for fitting a and b. This is clearly stated in the manuscript. We do not consider this as validation, but rather as check of the fit performance (the SD of this comparison is related to the RMS of the linear fit).

We have now also processed ECMWF profiles for 18 March and 18 September, covering the full seasonal cycle. Results are very similar to those from 18 December. For the uncertainty estimates for ECMWF, we now explicitly provide the numbers for 18 March 2018, where highest deviations were found.

For the regional model dataset, the description seems to indicate that it consists of 2 months (May and June 2018, see line 179), although it appears that only a few days (beginning of May) were used to derive the statistics. It would be advisable to use at least 2 months covering different seasons (so instead of May and June, maybe June and December).

The WRF simulations were performed by Vinod Kumar for a different purpose. While the full model simulation was set up for a 2 month period, however, only 9 days of simulations were available at the time of preparing the initial manuscript. Meanwhile, WRF simulations for the full period are available, so we applied the formalism to the full period. Resulting frequency distributions for $\delta_\Gamma$, however, did only change slightly.

Running the WRF simulations at high spatial resolution is computationally expensive. As different seasons are covered by ECMWF data, we do not see the need for an additional WRF simulation for winter.

For the third data set, data from radio sounding, I believe there are plenty of data available since meteorological services such as the MetOffice, launch weather balloons twice a day at several stations (I believe the DWD does the same).

In this study we focus on sonde measurements from the GRUAN network, which provides high consistency and thus good comparability. Though the number of stations is limited, and some stations only contribute only few profiles, the GRUAN dataset still covers a wide range of conditions (latitude, climate, altitude).

In total, we have now applied the O4 calculation to more than 200 million profiles.

The derived statistics are robust; errors of the mean and SD are negligible, and results are similar for ECMWF, WRF and GRUAN sites for quite different conditions.

Thus we would argue that the conclusions drawn in this study are supported by the presented data, and additional radio soundings are not required.

In a next step, these results need to be compared to a "current standard method" approximating O4 VCD. This is entirely missing in this manuscript.

For MAPA, the "current standard method" of determining the O4 VCD is based on ECMWF data. This was already included in Table 3.

But we agree that the comparison to currently used methods should be extended. Thus, we

1. added the following sentence to the introduction:

*... modelled profiles might not be available in some cases (e.g.~during measurement campaigns in remote regions and poor internet connection; for these cases, profiles from a climatology might be used as fallback option), ....*

2. added a new section (5.1) to the discussion, where the results for GRUAN profiles are compared to O4 columns from (a) daily model data, and (b) a climatology. In contrast to the discussion paper, we now correct for differences between surface altitudes from GRUAN vs. ECMWF, which has a large effect for mountain sites.

These comparisons indicate that the proposed calculation of the O4 VCD from surface values of p, T, and RH is indeed better than using profiles from a climatology.

I made a quick test using 3 years (2018, 2019 and 2020) of data from 6 stations (some overlap with stations that the authors use) from 12 UTC radiosonde launches (Cambourne [N=1061], Nottingham [N=315], Essen [N=468], Munich [N=1022], Lindenberg [N=1062], Lamont [N=430, 0 UTC for the latter to comply with the SZA requirement]). I find, using Eq.12 and Eq. 13, respectively, a high correlation (0.93, 0.94, 0.92, 0.93, 0.93, 0.93) for estimated and integrated O4 columns (confirming the authors' findings), and, using Eq. 15, low bias and low standard deviation (-1.8±1.0)%, (-1.5±1.1)%, (-1.0±1.3)%, (-1.7±1.1)%, (-1.4±1.2)%, (-0.8±1.7)% (again, confirming the authors' findings). If I do the same but, instead of using the approximation method, use climatology data, I get the following for mean bias and std: (1.6±2.0)%, (0.1±2.1)%, (0.2±2.1)%, (-

3.9±1.7)%, (0.8±2.0)%, (-1.2±2.3)% (so 50 -- 100% worse standard deviation, but partly better mean). However, the correlations are certainly much lower (0.45, 0.51, 0.55, 0.65, 0.61, 0.53). I would like to see similar comparisons in the manuscript in order to show that using this method is in fact better than using climatology values [Is it actually worth it? Please show this!].
It would be advisable to include such statistics for at least 20--30 other well distributed weather balloon launching sites for a few years.

We acknowledge that the reviewer applied the formalism to additional datasets, and see the consistent results as confirmation of our argument that the number of profiles presented in this study is sufficient in order to support the drawn conclusions.
We have added comparisons to O4 VCDs (including correlation coefficients) based on a profile climatology in section 5.1, and could indeed show that both correlation coefficients and SD are worse for the climatology-based VCDs.

Lastly, since the use of this approximation is clearly, as the authors point out, tailored towards parametrized MAX-DOAS inversion methods, it is highly desirable that the authors show some result of those: compare the retrieval results using the "previously standard method" to results obtained using this new method.

The study was indeed motivated from a parametrized MAX-DOAS perspective. However, the manuscript has a clear focus on the calculation of the O4 VCD. We would like to keep this focus, and we do not see the need for adding MAX-DOAS inversions to this study, as the proposed parameterization of the O4 VCD can be directly compared to the "true" values based on vertical integration. Thus, we just provide a rough estimate of the impact of changes of the a-priori O4 VCD on AODs derived with MAPA for the CINDI-2 campaign.

The manuscript frequently refers to Wagner et al. 2019 which makes it appear to be rather suited as an extension of that publication than a stand-alone publication.

Wagner et al., 2019, is indeed cited frequently, as it also deals with the calculation of the O4 VCD. However, the current study has a clear focus of parameterizing the O4 VCD by surface values of p, T, and RH alone, without constructing vertical profiles. This, with the completely new mathematical formalism and the extensive validation, we consider it to be appropriate for a stand-alone publication.

From my point of view, the innovative part of this manuscript is the empirical relation between the effective lapse rate and the surface relative humidity and the fact that this knowledge can be used in the formalism of Wagner et. al 2019 to replace the fixed lapse rate. This part however, I do find sufficiently important for publishing.
However, the current format of the manuscript is not good enough. It over complicates things. It does not include sufficient validation data, nor does it include a test on the final product this method is thought to be used for. Hence, I recommend intermediate revisions of this manuscript.
Restructure the manuscript. Streamline and simplify the method section, especially how to arrive at Eq. 9. For validation, extend regional model data covering a larger time period, use a separate day for the parameter fitting for the global model, use many more datasets from radiosondes. Include comparisons with currently used methods (such as using climatologies). Include tests showing that the final product (AOD retrieved from MAX-DOAS measurements) in fact benefits from this new method to estimate O4 VCDs by comparing results ("old" and "new") to complementary measurements.

We have revised the manuscript in response to the comments of both reviewers. We have restructured the manuscript, but still stick to the separation into section 2 on formalism and section 4 on applications in order to avoid forth-and-back jumps between formalism, results, and discussion. Thus, references to the "future" have been reduced, but cannot be avoided completely. We have extended the application of the formalism to additional WRF and ECMWF data. In addition, we have added a comparison to O4 VCDs derived from a climatology.

As argued above, we keep the derivation of Equations 1-6, which hold generally for any atmosphere, and derive Eq. 9 (now Eq. 10) later as special case for constant lapse rate.
The impact of a bias of the O4 VCD on MAX-DOAS profile retrievals is quantified in the introduction. We do not see the need for including additional MAX-DOAS profile inversions in this study, as the focus is set on the calculation of the O4 VCD.

**2. Specific comments**

- line 28: vertical profiles of T, P and RH are also needed to calculate the refraction index and hence are anyway needed for radiative transfer simulations, aren't they?

Within MAPA, the air mass factors used for the MAX-DOAS profile inversions are stored in a pre-calculated look up table based on RTM calculations using a standard atmosphere.

- line 33 -- 39: Here, the reader starts to wonder about the impact of temperature inversions. Later on (line 157), the authors mention temperature inversions and that they are far more frequent during night. Further, they mention that the main application is MAX-DOAS measurements and the presented method is limited to daytime. This should be already mentioned here, otherwise the reader will immediately wonder how temperature inversions are dealt with.

We added the following footnote to the introduction:

*Note that, for this approach, as well as for the parameterizations presented in this study, temperature inversions are problematic. As MAX-DOAS applications require daylight, however, night-time inversion layers are irrelevant for this study. The remaining temperature inversions at daytime, mostly occurring in early morning hours and over cold water and ice surfaces, will be discussed in Sect. 5.2.*

In addition, the impact of temperature inversions is now explicitly discussed in a new subsection 5.2 in the discussion.

- Section 2.1 seems largely unnecessary:
line 59 - 60: Incomplete list, and a 1 sentence paragraph: skip this.
line 61 -- 64: This should be skipped, does not add any new information (?) $n_{O4,0}$ == $n^2_{O2,0}$ is stated explicitly in Eq. 5
line 65 -- 68: No new information, all is contained in Table 1. Skip

We agree that Sect. 2.1 plus Table 1 adds some level of redundancy. However, we still consider it as helpful for the reader to also introduce the main quantities in the plain text. For instance, this allows to motivate the choice of units for O4 concentrations and column densities.

- Table 2: The g here corresponds to which latitude and which altitude? Since g varies about 0.5% between the poles and the equator, it should be important since the authors claim an accuracy of the same order of magnitude.

The value of $g=9.80665$ m/s$^2$ used in Metpy is the standard acceleration of gravity as listed in Tiesinga, Eite, Peter J. Mohr, David B. Newell, and Barry N. Taylor, 2020: The 2018 CODATA Recommended Values of the Fundamental Physical Constants (Web Version 8.1). Database developed by J. Baker, M. Douma, and S. Kotochigova. Available at https://physics.nist.gov/cuu/Constants/index.html, National Institute of Standards and Technology, Gaithersburg, MD 20899.
$g$ can be found on https://physics.nist.gov/cgi-bin/cuu/Value?gn|search_for=acceleration
Actual gravitational acceleration at equator and poles differ from this value by less then 0.27%. This is one order of magnitude lower than the critical uncertainty of about 3%. Thus we consider the effects of latitudinal changes of $g$ to be negligible. Note that also within ECMWF model data, the Earth is treated as sphere, using the same constant value of $g$:
https://confluence.ecmwf.int/display/CKB/ERA5%3A+data+documentation
In the revised manuscript, we added a footnote to $g$ in Table 2 stating that the effect of latitudinal dependence is within 0.27% and thus negligible.

With the precision for C given in units of K Pa-2 mol2 m-5, the value for C in units of K hPa-2 molecules2 cm-5 equates to 6.73267e39. The authors should consider to give one more digit of precision for the former units, so the unit conversion actually is consistent (a numeric value of 0.01856455 using the former units, which is consistent with the value given, indeed results [to the given precision] in the numeric value given using the latter units. However, as written now, it is not consistent). Is it really necessary to give the same constant in two different units?

In the revised manuscript, we provide only the second version of $C$, as this is the number needed in order to derive the $O_4$ VCD in the unit of $molecules^2/cm^5$ which is usually used in DOAS context.

• As mentioned above, I recommend to rewrite everything up to Equation 9 and derive this equation directly from the integration of the density assuming either a constant lapse rate (Eq. 9) or 0 lapse rate (Eq.8).

As argued above, we consider the formalism of equations 1 to 6 as an important part of this study, as this holds for general profiles. In addition, the effective heights of O2 and O4 are needed in order to calculate effective lapse rates.

However, if the authors really want to stick to their complicated way of showing things (I highly discourage this!!), they should consider the following points:
line 88 -- 89: Neither of equations 7 or 8 is based on Eq. 6. Equation 7 follows, as the authors write themselves, directly from the equation of hydrostatic equilibrium and the assumption of a constant O2 VMR (see comment below), Eq. 8 is simply the ideal gas law for quantities at the surface. Hence, this sentence should be moved to after line 100.

We do not claim that Eq. 7 or Eq. 8 is based on Eq. 6, but that the O4 VCD can be related to surface pressure, surface temperature, and lapse rate. This first sentence summarizes the formalism derived in section 2.3 in plain words, and we think that it makes sense to start the section with this sentence.

Eq. 7: While the authors spoon feed the derivation of Eq. 6, they skip totally the derivation of Eq. 7 from the equation of hydrostatic equilibrium  dp = -rho g dz. (e.g. by Ptop - Pbottom =  - rho g h - (Ptop=0)--> -P0 = - rho g h --(rho = n*m/vol)--> P0/g = n m h / vol  --(n h / vol == Vair)--> P0/ g/ m = Vair). Apart from assuming that, it is also rightfully assumed that the volume mixing ratio is constant; although implicitly clear by Table 2, it should be added.

The surface pressure results from the total weight of the air mass above, which is directly proportional to the $O_2$ VCD. Thus, for Eq. 7, it is not necessary to integrate the hydrostatic equilibrium.
We have modified the paragraph to:
*Assuming a hydrostatic atmosphere, the surface pressure is just the gravitational force per area of the total air column.*
*Thus, the O2 VCD is directly related to the surface pressure: ...*
We do not think that it is necessary to explicitly mention that the O2 VMR is constant in the text, as it is listed in the table of constants.

line 98 -- 101 (+ Appendix A): All equations should have a number. Please add a number to the equation stated here. Also,reference the derivation of this directly as appendix A2.

The equations given in the text are just summing up the results of the derivation in Appendix A. Giving them a number would result in having the same equation twice with different number.
In the revised manuscript, we clarify this by adding references to the respective equation numbers in Appendix A.

Regarding equation A3 from the appendix A2: Also here, a derivation of this equation is missing. Maybe start with the equation for  polytropic atmosphere p/p0 = ( T/ T0)**(gM/R/Gamma)   and T = T0 + Gamma * z' and say that Eq. A3 follows from this together with the ideal gas law.

We consider Eq. A3 as text book knowledge that does not need a derivation; for instance, it is provided on Wikipedia (https://en.wikipedia.org/wiki/Barometric_formula)

I think there is something not going quite right with the signs: The authors define the laps rate to be negative. As such (since g is defined positive in Table 2), -alpha +1 = -gM/R/ Gamma would be (Gamma < 0) a positive quantity. With this, the term in round brackets in Eq. A6 for z' = ∞ would otherwise not disappear. I also wonder if there might be a minus sign missing in Eq. A6 in the last line (or z should be maybe going from 0 to -∞, this solves the problems, I think, please check) ....

We thank the reviewer for checking the formalism in such a detail and pointing out to an error in Eq. A6: indeed, the term in round brackets does not disappear for z' = ∞. However, this is not related to the signs, which we have checked to be correct.

We have investigated this in further detail and finally noticed, that a profile with a constant lapse rate cannot be extended to infinity, but has to end at an altitude of $z_{TOA}$ = -T0/Gamma, which is about 46 km for T0=300 K and Gamma = -6.5 K/km. Above this altitude, T would be negative, and the number concentration given by equation A3 would be a complex number, which is both unphysical.

So the integration in equation A6 has to be performed from 0 to $z_{TOA}$.

In this case, the first term (z=$z_{TOA}$) now vanishes, as the term in round brackets becomes 0, and the remaining equations A7-A9 are still correct.

We have modified this accordingly in the revised manuscript.

In line 429, what does "12" refer to? Eq. 12? It's easy to make mistakes, so it is of course also possible that I made a mistake in my checking. Please carefully check the signs anyway.

"12" refers to Eq. 12. The respective paragraph (now a new subsection in Appendix A) was revised accordingly.

- line 116 -- 128: The authors refer to a section "in the future". This is very bad practice. Reorganize the structure of the article in order to make it possible for the reader to follow.

See general reply (*) above (top of page 2).

- line 117 -- 118: Please explain this statement.

This section has been largely revised, and the statement has been skipped.

- line 118 -- 119: Please refer to the equation where you show that this statement is true. "effective" laps rate is in fact not defined, what makes the lapse rate an "effective" one?

In the revised manuscript, we define the effective lapse rate directly in subsection 2.3.

- line 127 -- 128: Please explain what you mean by this statement. Why "basically"?

We have largely extended the description of the fit of parameters a and b and reformulated and clarified this statement.

- line 130: The authors refer to "the future". Please restructure the manuscript.

We removed the reference to "the future" here as it is not necessary for the understanding of this section.

- line 163/ line 167: what does "truncation at T639"/ "truncation at T255" mean?

As explained in ECMWF FAQs, "the IFS model uses a spherical harmonic expansion of fields, truncated at a particular wave number. For example T1279 identifies truncation at wave number 1279. Each spectral truncation is related to a regular Gaussian grid, which is regular in longitude and almost regular in latitude."

https://confluence.ecmwf.int/display/UDOC/What+is+the+connection+between+the+spectral+truncation+and+the+Gaussian+grids+-+Metview+FAQ

In this manuscript, however, we do not want to go in such detail. We modified the text to *"truncated at wavenumber 639/255"*, and consider this information to be helpful for readers familiar with atmospheric models, while others could just read on.

- line 165: The authors are using one and the same day for fitting the parameters and for validation. This has to be changed.

We are aware that the 18 June cannot be used for independent evaluation of the parameterization performance, as the parameters were fitted for this day. This is also clearly stated in the manuscript. We now provide ECMWF uncertainties for a different day (18 March 2018) in the text.

- line 168: How is this a "pre-processed" data set if the authors use the model out put and post-process it?

This processing was made by ECMWF, so from our perspective, it is a pre-processed data set.

- line 195: Are the authors following some recommendations with this? If so, please cite. Otherwise, please explain why this choice justifies the statement "only retain the measurements of the highest quality".

In order to avoid a too detailed description in the main manuscript, we have moved this section to the appendix. We have modified the respective paragraph as follows:
*Additionally we have applied quality control filters such that the parameters QUALITAETS_BYTE (QB) is below 4 (thereby excluding untested, objected, and calculated values), and QUALITAETS_NIVEAU (QN) is either 3 (automatic control and correction) or 7 (second control done, before correction) to only retain measurements of high quality; note that using only data with QN=10, as recommended in the DWD description, would result in having almost no data left. Additionally, we have applied quality control filters such that the parameters QUALITAETS_BYTE (QB) is below 4 (thereby excluding untested, objected, and calculated values), and QUALITAETS_NIVEAU (QN) is either 3 (automatic control and correction) or 7 (second control done, before correction) to only retain measurements of high quality. By applying these criteria, we retained 98.2%, 100%, and 99.5% of T0, p0, and RH0 data, respectively. Note that using only data with QN=10 (the best possible quality check level) would result in no data left for the period considered in this study. If only QN=7 had been applied, we would have retained the same number of T0 and RH0 but no p0 data.*

- Fig. 1: Can the authors comment on the apparent differences between the models and the GRUAN measurements about the covered parameter space? The author mention the very high values, but do not comment on the many missing intermediate values.

We have modified the figure caption as follows:
Low values correspond to high altitude sites with low surface pressure; for GRUAN, only few of such stations are available (Boulder and La Reunion at 1.7 km and 2.2 km altitude, respectively).

The authors use 2 months of WRF data but choose to show only a single day (in a). The authors say that this is in order to keep the figure readable. I suggest to include a second row showing all the data. From statements elsewhere in the manuscript, I have the impression that even the data used for the statistics is not the full 2 months, but only a few days from the beginning of May. Is this correct, if so, why?

- line 217 - 218: Please show this in a plot (see comment above, include all data points).

In the revised manuscript, we now present 2d histograms instead of scatterplots, and include the complete 2-month WRF data set.

- Sect. 3.2: The little detail given is really not well structured. I suggest to include a thorough description as an appendix.

As proposed by the reviewer, we have moved this paragraph to the appendix, and slightly modified its structure.

- line 219 -- 220: This statement is not quite correct. In terms of covered V, both WRF and GRUAN cover roughly $\hat{a}^{\wedge}\dagger V = 0.7e43$ molec$^2$/cm$^5$, where WRF covers the space more evenly, GRUAN has a gap of $\hat{a}^{\wedge}\dagger V = 0.2e43$ molec$^2$/cm$^5$.

The statement was not referring to the variability of the O4 VCD, but to the variability of slopes (corresponding to different lapse rates). We have modified this sentence to
*ECMWF and GRUAN data show higher variability in slopes …*

- line 225: Where do you show that this statement is correct? There is no direct comparison made. Either include such a comparison or remove this statement.
- line 229: Please show this 0.5% explicitly.

In the manuscript draft, we have discussed and quantified the difference between Eq. 9 and the approach proposed in Wagner et al. in Appendix A2 (lines 425-429).
In the revised manuscript, we extended this discussion in a new subsection of (Appendix A3) and explicitly calculate the effect of neglecting the tropopause on the O4 VCD.

- Figure 2: Why is the histogram only considering 8 days instead of the complete 2 month data set? Or do I understand this incorrectly?

In the revised manuscript, results from the full 2-month period are included in the WRF histograms. The resulting frequency distributions of $\delta_\Gamma$ and $\delta_{RH}$, however, changed only slightly.

- Figure 3: Maybe comment on great lakes in North America in summer.

We would like to thank the reviewer for pointing this out. We had a closer look at the profiles above the Great lakes and noticed a strong near-surface temperature inversion due to the cold water. Similar effects are visible over the oceans where SST is low.
We have added the following sentence to the manuscript (at line 237, after "High values … over ocean."):
*In particular over cold water surfaces, like the West coast of North and South America, the Hudson Bay, or the Great lakes, $\delta_\Gamma$ is very high (up to 7%). This is related to temperature inversions close to ground: due to the too low surface temperatures, the O4 VCD calculated from Eq. 10 is biased high.*
In addition, we have added a new Figure (Fig. 9) showing maps of temperature gradients close to the ground, and discuss the impact of temperature inversions in a new subsection (5.2) in the discussions.

- line 238 - 239: You do not show this. Either show it (appendix?) or remove this comment.

We removed this comment.

- line 239 - 240: Can you prove this statement? Please show a plot of lapse rate vs. $\delta_\Gamma$

In the revised manuscript, we have added a new figure, showing maps of the effective lapse rate for ECMWF data on 18 June 2018, supporting our statement.

- line 242: How is this considered (implicitly due to known profiles)?

We have specified the description about how the true O4 VCD is derived as follows:
*… we also calculate the "true"' O4 VCD, which is derived by*
*(a) calculating the profile of n_O2 from profiles of T, p and RH. In this step, the effect of humidity is explicitly accounted for by subtracting the water vapor pressure before calculating n_O2 based on the ideal gas law. …*

Also include a reference and an equation using the partial pressures of dry air and water vapour which makes the dependence on specific humidity apparent.

- line 243: Please define specific humidity (mass ratio of water vapour content and total mass of air parcel) and relate it to relative humidity (ratio of water vapour pressure and equilibrium water vapour pressure).

We have modified the line of arguments concerning humidity effects. We directly introduce and motivate the parameterization of effective lapse rate and thus the O4 VCD based on RH_0 in Sect. 2.

The effects of humidity on O4 number density are also (at least partly) covered by the empirical fit of a and b, and the impact of specific humidity turned out to be not critical for the proposed parameterization, as we now demonstrate in a new subsection in the discussions, where we directly check the relation between $\delta_{RH}$ vs. specific humidity and TCWV.

Since no significant effects were found, we would like to keep this discussion on a more general level, and we would prefer not to add additional formulae which are irrelevant for this study.

- Sect. 4.2.: The authors lack to clearly state the logical chain of causes here: relative humidity affects effective lapse rate. Lapse rate affects V.

We have strengthened the logical chain accordingly.

- line 250 - 256: Make a plot as Fig. 3 using effective lapse rate calculated from the ECMWF model to show that this statement is correct.

In the revised manuscript, we have added a new figure, showing maps of the effective lapse rate for ECMWF data on 18 June 2018. In addition, we provide 2d histograms of effective lapse rate versus RH0.

- Sect. 4.3.:
It is stated that ECMWF data from June, 18th 2018 was used for the fit. Further, the authors state (line 267) that they investigate June, 18th 2018. This should never be done. You cannot use the same day for fitting and verification. Please choose a third day for the fitting and use the same day (June 18 and December 18) only for verification.

We are aware that the 18 June cannot be used for independent evaluation of the parameterization performance, as the parameters were fitted for this day. This is also clearly stated in the manuscript. In the revised manuscript, we provide results for 18 March for evaluating accuracy and precision.

- line 264: Please state clearly which figures to compare. Also, it seems that for certain regions (e.g. the Andes, central Europe around lunch time), the absolute value of $\delta_\Gamma$ increased. It might make sense to show a map of the relative improvements of $\delta_\Gamma$. (? or maybe not...)

We added references to the figures showing $\delta_\Gamma$ and $\delta_{RH}$ for WRF and ECMWF. There are actually few regions where $\delta_\Gamma$ is closer to 0 than $\delta_{RH}$, but this is no contradiction to the frequency distributions of $\delta_\Gamma$ and $\delta_{RH}$.

Relative improvements would require divisions by small numbers, as $\delta_\Gamma$ and $\delta_{RH}$ are close to 0. This would lead to instabilities.

- Table 3: Include, in analogy to Eq. 14 and 15, an equation for $\delta\_ECMWF$. Include also a histogram for the data in Table 3 in analogy to the histograms in Figs. 2,3,4,6,7,8. Why do the authors not present a correlation plot of V true and V parametrized? I think this is could be very instructive (I made it for the aforementioned stations and it looks very nice).

Instead of including a further definition for $\delta\_ECMWF$, we now just define the deviation in a generic way for $\delta\_x$, where x can stand for $\Gamma$, RH, or ECMWF.

We do not present GRUAN results in form of a histogram, as the separation for different stations would be lost. Instead, we now present GRUAN results in a new figure (Fig. 8) which also includes the comparison to O4 VCDs based on daily as well as climatological ECMWF profiles. We also

included the respective correlation coefficients to this figure. We don't consider additional scatterplots necessary here.

- line 301: with "radiation shield", you mean a Stevenson screen?

Direct solar radiation must be shielded from the thermometer, otherwise the measurements cannot be used. Stevenson screens are used for stationary meteorological stations. But there are also small, portable devices available with integrated radiation shields, which might be easily installed next to a MAX-DOAS instrument. So we keep the sentence as is.

- line 303: What is sufficient?

As stated in the introduction, the uncertainty of the O4 VCD should be below 3%. In order to make this more clear, we extended the respective sentence in the introduction to
*Thus, for MAX-DOAS profile inversions, the O4 VCD should be determined with accuracy and precision better than about 3%, which limits the impact on resulting AODs to below 10% and leaves other sources of uncertainty, i.e. the spectral analysis ($\approx$ 5%) as well as radiative transfer modeling ($\approx$ 4%) (see Wagner et al., 2021, Table 3 therein) as the limiting factors.*

- line 319 - 320: Please explain what you mean by "and $V_{O4}$, RH is almost the same for WRF and ground stations".

We meant that the diurnal cycles of $V_{O4}$, RH is almost the same for surface data from WRF vs. DWD. This statement is not as clear for the revised results based on 2 months of WRF simulations. We have revised the paragraph accordingly.

- line 334: What is sufficiently here? Why do you judge it to be sufficiently?

We have modified this statement to
*Thus, the parameterization of eq. 14 also reflects most of the diurnal cycle of the O4 VCD, with remaining systematic errors below 0.3%.*

- Figure 10: How did you choose the points to be plotted? Are the correlation values indicated still using all points?

For the calculation of correlation coefficients, all data points were used. The scatter plots showed a subset by just selecting pixels in steps of 100 in order to keep them readable.
In the revised manuscript, we replaced all scatterplots by color-coded 2d histograms based on the complete data set.

- line 354: Where does the 3% estimate come from?

In the revised manuscript, we provide concrete numbers for $\delta_{RH}$ for elevated sites, which are actually lower than 3% after skipping profiles affected by temperature inversions.

- Sect. 5.5: How do you measure $S_{O2}$?

In response to the comments of reviewer 2, we have moved this section into the Appendix (App. A4). In the revised manuscript, we modified this section and now only refer to the VCDs of $O_2$ and $O_4$.

- line 379: Why "basically"?

We have removed "basically".

- line 386: Why is it sufficient?

We have extended this sentence to
*This accuracy and precision of < 3% is typically lower than other uncertainties of spectral analysis or radiative transfer modelling (Wagner et al., 2019).*

- line 389: Inside a Stevenson screen I assume, otherwise the readings might be rather useless.

See reply to line 301 above. We modified the sentence to

*... state-of-the-art thermometer (with radiation shield), barometer, and hygrometer.*

**3. Technical comments**

3.1 general

Since one of the co-authors is the chief-executive editor of AMT and the first author is an associate editor, I find it slightly worrying that the authors disregard so many of the AMT guidelines:

We thank the reviewer for listing the inconsistencies to AMT guidelines. In the revised manuscript, we have resolved most of the raised issues. In addition, Copernicus office will take care of consistent style and format during the copyediting process.

- The journal guidelines clearly state that the recommendations of the SI brochure and the IUPAX Green Book (links can be found here: https://www.atmospheric-measurement-techniques.net/submission.html#math) should be followed. This is largely neglected in the axis labels and table headers. Physical quantities and units should not be written as "quantity [unit]" but as "quantity/ unit". Consider SI brochure Sect. 5.4 or alternatively, page 3 of the IUPAC Green Book. Please adjust this throughout the manuscript.

As noted by the reviewer, this is a recommendation, but not a strict standard. Thus we prefer to provide units in brackets in figure axis, which is also commonly done in most of recently published AMT papers.

- The journal guidelines clearly state that universal time should be indicated as "UTC". Please correct all "utc" (e.g. Figure 9) in the manuscript.

Done.

- The journal guidelines clearly state (https://www.atmospheric-measurement-techniques.net/submission.html#figurestables) that table should be written Table if followed by a number, please correct throughout the manuscript (e.g. line 26, 263, 283, 306).

Done.

- The journal guidelines state: "Coordinates need a degree sign and a space when naming the direction (e.g. 30° N, 25° E)". This is not done anywhere, please correct throughout the manuscript (e.g. in Fig. 2, 6, B1 and C1, in Table E1, line 175 no space is included).

Done.

- Inconsistent use of section (most of the manuscript) vs sect. (line 279, 217). Journal guidelines say "Sect." unless at the start of the sentence. Please correct.

Done.

- Inconsistent use of fig. (e.g. line 177, 197), Fig. (e.g. 216) and Figure (338). Journal guidelines say "Fig." unless at the start of the sentence. Please correct.

Done.

- Inconsistent use of equation (e.g. line 360) and eq. (Table 2, line 83, 88,...) Journal guidelines say "Eq." unless at the start of the sentence. Please correct.

Done.

- The journal guidelines clearly state not to use hyphens for ranges, but to use en dashes to indicate ranges (https://www.atmospheric-measurement-techniques.net/submission.html#english). However, the authors use sometimes hyphens (e.g. Fig.9 caption). Other times they use "to" which,

according to https://www.atmospheric-measurement-techniques.net/submission.html#math is ok. Please change the hyphens to en dashes or "to".

We use "to" for indicating ranges in the revised manuscript.

- "data" should be considered plural (https://www.atmospheric-measurement-techniques.net/submission.html#english). Please correct throughout the manuscript. (e.g. 253, 169, 460)

Done.

3.2 specific
- line 14: What is absorbed is the light, not the O4, hence "O4 absorption in scattered light" is not a correct formulation. (Maybe add "pattern"?)

We have added "pattern".

- line 15: "light path distributions in the atmosphere" seems also not quite correct.

We do not see a problem here.

- line 15: "light path increases" should be "light path length increases"

Done

- line 16: "cloud heights": be more specific: cloud top heights or cloud base heights?

As the $O_4$ signal is caused by a complex combination of light path shortening or lengthening and multiple scattering inside the cloud, it is neither top nor bottom, but rather something in between. As this is not the focus of this study, and a more specific statement would require additional explanations distracting from the main line, we would prefer to keep this statement unspecific here.

- line 29: "[...] measured profiles [...] do not provide continuous temporal coverage [...]" The profiles do not provide temporal coverage of what? (of itself?) This a somewhat awkward formulation, please reformulate

We changed the subject to "radiosonde measurements".

- Table 1: "Relative deviation between of parametrized and true O4 VCD" --> remove "of"

Done.

- Table 2: replace e+39 by "$\times 10^{39}$ "

Done.

- line 70 -- 82: Why so wordy? Eq. (1) -- (3) can be summarized in one line of equation without the unnecessary text.

We consider it necessary to introduce equations 1 to 3 with some level of detail, even if the single steps seem trivial. In particular for the definition of the effective lapse rate, the concept of the effective height is required.

- line 86: add coma: "So far, .."

Done.

- line 117: "on first glance" --> "at first glance"

Done.

- Eq. 12: inconsistent accuracy of C (c.f. Table 3).

In Table 3, constants are now provided with high and consistent accuracy. For the final equation given in the plain text, which should be applied by the user, less digits are sufficient, given the remaining uncertainties.

- line 156: add "profiles" after "daytime".

Done.

- line 159: "selecting data for ..." wrong preposition.

We replaced the preposition by "with".

- line 164: add a coma after "here"

Done.

- line 177: Insert coma after "Vertically"

Done.

- line 180: Reverse the sentence. Start with "For constraining.... we use..." otherwise it is confusing why you start again with ERA5 data.

Done.

- line 180: include "of" after resolution

Done.

- line 184: "The selection of SZA < 85°" is not correct. You do not select the SZA, you select data at times of the day at which SZA < 85°. Please reformulate.

We modified this sentence to

*The selection of data with SZA < 85° ...*

- line 185: replace "reach up to a pressure level" by "extend to a pressure of" or similar.

Done.

- line 211: "if" --> "of"?

Corrected.

- line 212: Please reformulate this sentence (especially "apply eq. 9 in section...")

We have reformulated this sentence.

- Sect. 4.1. title: add "a" before function.

Done.

- Fig. 1: axis labels are too small. Figures are too small, extend to page width. Legend box partly covers line. "mountaineous" --> "mountainous" or better: high altitude. If the authors choose to use only y-tick labels and y labels on the first subplot, the hspace should be 0.

We have increased font size for axis labels. The figure width was on purpose chosen as ¾ of page width, in order to have the same subpanel size for figures 1, 5, 10, and D1 (now: 1, 5, 10, 11, B3). The legend box was adjusted. We modified the caption to "high altitude". y-tick labels are now shown for all subplots.

- line 218: add "the" in front of "highest"

Done.

- line 219: "matching to a lapse rate" needs reformulation

We have replaced "matching" by "in accordance".

- line 230: The choice of $\delta_\Gamma$ is not the best, it seems to indicate that the $\delta$ is w.r.t $\Gamma$ while it is w.r.t. V. Please consider renaming. Is it really "parametrized" yet? As I understand, the authors apply Eq. 9 with the constant lapse rate. So I do not see any parametrization here.

$\delta$ is used as symbol for relative deviations between calculated and "true" O4 VCDs. So both $\delta_\Gamma$ and $\delta_{RH}$ refer to differences in V. But we tried to discriminate the results for the two investigated approaches by a clear, but short subscript. We think that "$\Gamma$" (using the lapse rate as parameter, thus "parametrized") and "RH" (using RH at ground as parameter) fulfil this purpose.

- Figure 2: Please repeat the meaning of $\sigma$ and $\mu$ (from line 147) in the figure caption. Swap color bar and histogram, include ticks on the right hand side of the histogram. Please consider putting the coordinates at the axis instead of in the middle of the figure (same for Fig. 4).

We do not use $\sigma$ and $\mu$ any more in the revised manuscript. We would like to keep the arrangement of the subpanels as is in order to have the color bar next to the maps it refers to. Putting the lat/lon coordinates at the axis would need additional space outside the maps, and consequently further shrink down the maps in the subpanels. Thus we would prefer to have the lat/lon coordinates inside the figure.

- line 235: "Also for ECMWF data..." should really be something like "Considering ECMWF data as the basis for calculating $\delta_\Gamma$, also results in $\delta_\Gamma$ values close to 0 for the area covering Germany". Please reformulate.

We have modified this sentence to
*For ECMWF data on 18 June 2018, $\delta_\Gamma$ over Germany is close to 0 as well.*

- line 237: "For continents..."needs reformulation.

We have modified this sentence to
*Over continents, $\delta_\Gamma$ is lower, and generally close to 0, except over deserts, where negative values are observed.*

- line 238: It is advisable to stick to either abbreviations or the symbolic notation, do not mix.

We use the terms mean and SD throughout the revised manuscript, and skip $\mu$ and $\sigma$.

- line 239: insert "the" in front of "same".

Done.

- Figure 3: Add year after "18 June" (or reformulate to "the same day").

Done.

- Figure 4: Add year after "18 December" (or reformulate, see above).

Done.

- line 242: I think it is more correct to refer to the density instead of the weight of air. Please reformulate.

We now state that addition of humidity reduces the O2 number density.

- line 244: Is "compared" really the correct verb to use here?

As we show both quantities in a scatterplot (now 2d histogram), we think that we "compared" the quantities.

- line 252: "Subsidence" of what?

We have extended the discussion of the relation of RH0 and effective lapse rate in section 2.5, including the effect of "large-scale subsidence of air masses". The text in line 252 has been skipped.

- line 257 -- 259: Please reformulate and clearly state that you used Eq. 11 to fit parameters a and b and the result is Eq. 12. As it currently reads, it is hardly comprehensible.

We have largely revised and extended the description of how the linear fit is performed to gain a and b, including a new figure showing the fitted line.

- line 270: "low values are improved"? Reformulate, e.g. "Areas where $\delta_\Gamma$ showed large negative values, show less extreme $\delta_{RH}$".

We modified this sentence to

*The large difference between deserts and oceans seen in $\delta_\Gamma$ (Fig. 3) is strongly reduced for $\delta_{RH}$ (Fig. 7).*

- line 274: Why "basically"?

We skipped "basically".

- line 276: "weather condition [...] are usually not considered in MAX-DOAS retrieval". This sentence does not make sense, what the authors want to say is that days with such weather conditions are usually not considered for MAX-DOAS retrieval.

We have re-formulated the sentence to

*Note, however, that MAX-DOAS retrievals are usually not considered for weather conditions associated with rain and clouds.*

- line 296: I think that the 3% is not supported by the plots Fig. 8 (Fig. 7 has to be disregarded because that day was used to fit the parameters). Values of up to ±6% seem more correct, but it is hard to see from the figures.

Most of the large positive deviations in Fig. 7 and 8 are related to temperature inversions, which is discussed in detail in the revised manuscript. For the discussion of possible effects of z0, we have thus skipped profiles affected by temperature inversions. The remaining profiles with z0>2 km reveal a mean deviation of -0.5% with a SD of 1.8%.

- line 300: this seems to have the wrong indentation, please check.

Lines 300 to 302 are part of the enumerate block starting at line 297, so the indentation is correct.

- line 315: add a comma after "this"

Done.

- line 317: "for" --> "at"

Done.

- line 321: Do not start sentences with "But"

We would like to ask the Copernicus language editor for a suggestion how to formulate this sentence.

- line 320 - 320: insert "order of" between "same" and "magnitude".

Done.

- line 320 - 321: This sentence is incomprehensible

We have largely revised the respective paragraph.

- Figure 9: Axis labels too small. "[...] all cycles are referred to the mean value [...]". This sentence does not make sense. Please reformulate.

We have increased the complete figure in the revised manuscript and modified the figure caption to
*For better comparison, the mean value at 11:00 UTC (around solar noon for Germany) is subtracted from all datasets.*

- Figure 10: Axis labels and Axes tick labels are far too small. Please adjust the range of the x-axis of panel (c). Please check the y-axis label: Should this not be $\delta_{RH}$?

We have increased font size for axis labels. The figure width was on purpose chosen as ¾ of page width, in order to have the same subpanel size for figures 1, 5, 10, and D1 (now 1, 5, 10, 11, B3). The range of x in panel (c) was on purpose set to the same range as for (a) and (b) for better comparability. We have corrected the label of the y-axis.
In addition, we now show density plots instead of scatter plots in order to include the complete datasets in the panels.

- line 368: refer to Eq. A9 instead of Appendix A.

We moved this section to Appendix A4 and added references to the respective equations.

- Appendix B and Appendix C are never mentioned in the text.

In the revised manuscript, the Appendices about data sets have been extended, and they are now mentioned in the main text.

- line 387: remove comma after "measurements".

This paragraph has been largely revised due the addition of profiles from a climatology.

- Figure D1: axes labels are not readable. "DWD--> " and "WRF -->" seem to indicate a direction as displayed. Maybe remove the arrow, it is misleading.

We have increased font size for axis labels. We have removed the arrows from DWD and WRF. In addition, we now show density plots instead of scatter plots in order to include the complete datasets in the panels.

- line 442: add comma after "maintain".

Done.

- line 443 - 444: "This is a consequence of the spatial resolution of the WRF simulations of 1 km not resolving single mountains". This sentence seems incorrect. Reformulate.

We have re-formulated this sentence to
*This is a consequence of the spatial resolution of the WRF simulations of 1 km, which is not sufficient for resolving single mountains.*

- Figure E1: station labels are not very well readable in the figure.

We have increased the font size of station labels in Fig. E1 in the revised manuscript.

---

## Author Comment (AC2)

**Corrigendum**

We would like to thank both reviewers for the constructive feedback to our study, which helped to improve the paper. In particular, while preparing the replies to the reviewers, we noticed two bugs in the data presented in the AMTD paper:

1. As suggested by reviewer #1, we have checked the correlations of O4 VCDs based on GRUAN and interpolated ECMWF profiles. This comparison revealed that the time axis was flipped in the ECMWF data (while the intention was to flip the altitude axis only). I.e. the standard deviations for ECMWF listed in table 3 were too high, while the mean values were almost unaffected. We have corrected this in the revised manuscript. In addition, we now also correct for differences between surface altitudes from GRUAN vs. ECMWF, which has a large effect for mountain sites. The numbers for $\delta_{\text{ECMWF}}$ have thus changed (generally improved) considerably.

2. As proposed by reviewer #2, we have added information on the time coverage of the analysed GRUAN sonde launches. By comparison with Table E1, we then noticed that the number of sonde launches did not match. Actually, Table E1 listed the number of *all* GRUAN profiles rather than just those for SZA < 85°. We have corrected this in the revised manuscript.

In addition, we noticed a further necessary modification:

3. As the regular latitude-longitude grid of the global ECMWF data over-represents high latitudes, we now only consider the fraction of pixels corresponding to cos(lat) for each latitude for the calculation of histograms, means and standard deviations. This also affects the determination of fit parameters $a$ and $b$, which have slightly changed. Consequently, all derived numbers that are based on $a$ and $b$ had to be updated accordingly. However, changes are in the permil range, and the general findings and conclusions of this study did not change.

---

## Author Comment (AC3)

**Reply to reviewer #2**

We provide a point-by-point reply to the issues raised by reviewer 2 below. The original review is included in grey. Text changes in the manuscript are indicated in italic font.

Please also note the modifications made in the revised manuscript that are specified in the Corrigendum.

The manuscript by Beirle et al. presents a parameterization of $O_4$ vertical column densities (VCD) based on surface observations of temperature (T), pressure (p), and, ultimately, relative humidity (RH). A first parameterization that only consider p and T is derived based on first principles and performs reasonably well when compared to "true" $O_4$ VCDs calculated from WRF, ECMWF, and radiosonde data. The authors use a modified version of the first-principles parameterization and the true $O_4$ VCD to develop an empirical parameterization that also include surface RH. This empirical parameterization improved the $O_4$ VCD calculations to below a 2% uncertainty that is needed for MAX-DOAS based inversions. The authors identify several instances in which the parameterization is less accurate, such as condition with surface inversions and mountainous regions.
Overall, this is a well-written manuscript that present a new method to improve $O_4$ VCD calculations. The presented parameterizations will be useful to the MAX-DOAS community, which needs these VCDs for their retrievals. There are some parts of the manuscript that could be further strengthened, as I will outline below, and a few minor text/language issues. The manuscript fits well into AMT, and I recommend its publication after minor revisions.

We thank the reviewer for the positive assessment. Below we reply to the raised issues point by point.

Detailed comments:
Line 44-45: It seems unlikely that a lapse rates close to 0 can be achieved due to condensation alone. There likely some dynamic reason as well. It may be worth citing books/manuscripts from the meteorological literature that give an overview of potential atmospheric lapse rates here.

This is a misunderstanding; we do not want to claim that the lapse rate becomes 0 by condensation. Instead, the addendum "closer to 0" is meant to indicate the direction of the change; as the dry lapse rate is negative, it is lower than the moist lapse rate, but its absolute is higher. In order to avoid confusion, we do not use "lower/higher" here. We modified the respective sentence as follows and hope that this avoids misunderstandings:
*... parts of the oceans with weaker (i.e. closer to zero) lapse rates due to condensation.*

Section 3.4: Please provide some more information on the time frame over which the sondes were flown. It also seems that some of the locations had very few sondes, thus making the statistical interpretation challenging. It may also be a good idea to add the number of sondes for each location to Table 3.

We thank the reviewer for this comment. We have added information on the time coverage of the analysed sonde launches to Appendix E. By comparison with table E1, we then noticed that the number of sonde launches did not match. Actually, table E1 lists the number of all GRUAN profiles rather than just those for SZA < 85°. We have corrected this in the revised manuscript.
We agree that for several stations, statistics are quite limited (even more for the corrected number of available profiles). However, if stations with few profiles would be skipped, some conditions (e.g. tropics) would not be included any more. We still consider the limited information content of these stations to be valuable for this study, as none of the stations shows any exceptional behaviour.
In response to the comments of reviewer 1, we have decided to present the data of table 3 in a new figure in the revised document. In order to indicate low statistics, the results for stations with few profiles are marked by lighter color.

Section 5.4 and 5.5: These sections present some interesting ideas. However, the proposed formulas are not backed up by any data or detailed analysis. I also found these sections rather distracting from the main point of the paper. They should either be expanded by showing that the calculation of lapse rates yields reasonable results by comparing them to the meteorological data the authors have already used in the manuscript or, which would be my recommendation, be moved into their own publication.

We understand that sections 5.4 and 5.5 of the discussion paper could be considered distracting. We have thus revised the manuscript as follows:
- Concerning Sect. 5.4, the effective lapse rate is now already defined in the formalism section (2.3). We now also present a comparison of the effective lapse rate to the 5 km lapse rate from ECMWF profiles in Sect. 4.
- Concerning Sect. 5.5, this is so far not more than an idea for a future application, which might indeed become a separate publication as soon as substantiated by measurements. Nevertheless, we would like to mention this idea already in this manuscript. In order not to distract the logical flow of the discussion, we moved this subsection into a new subsection of Appendix A, where the ratio of effective heights is discussed.

Lines 257 – 259: This is such a central part of the manuscript that I would recommend expanding it to provide the reader with more information on how the parameters a and b were derived. Maybe add a figure of the data and the fitting line. In addition, please provide uncertainties and $R^2$ of the fit.

We agree that the description of the fit was lacking for detail.
In the revised manuscript, we have clarified the fitting procedure in section 2.5 (formalism).
The fit parameters (with uncertainties) are now derived in section 4.2, which also includes a new figure showing the data, correlation coefficient and the fitted line.

Line 329 – 330 (and other places in manuscript): I believe this could be generalized in stating that the parameterization loses accuracy when surface temperature inversions are present, i.e. in the morning and evening, in the Arctic, etc.

We agree. In the revised manuscript, we have added the following footnote to the introduction:
*Note that, for this approach, as well as for the parameterizations presented in this study, temperature inversions are problematic. As MAX-DOAS applications require daylight, however, night-time inversion layers are irrelevant for this study. The remaining temperature inversions at daytime, mostly occurring in early morning hours and over cold water and ice surfaces, will be discussed in Sect. 5.2.*
We discuss the effect of temperature inversions in detail in a new section (5.2) in the discussion, including a map of surface temperature inversions in ECMW data on 18 June 2018 that clearly illustrates that for these conditions higher deviations are found.

Line 39: "… as the main source…"
Done.

Line 42: "The main reason…"
Done.

Line 159: introduce SZA here by spelling out "solar zenith angle"
Done.

Line 350-351: change to "Obviously, other factors would probably also have to be…."
Done.

---

## Author Response (AR2)

Mainz, 3 January 2022

Dear Michel van Roozendael,

please find a revised version of the manuscript amt-2021-213 according to the minor revisions suggested by the reviewer.

Concerning the main issues:

(1) re-include datasets with temperature inversion (during day time)
This seems to be a misunderstanding: daytime temperature inversions are not skipped from the analysis: They are included in the global maps as well as in the presented histograms and the listed standard deviations.
Only for the investigation of possible dependencies on humidity (Fig. 10) and surface altitude (Fig. 11) we had to skip temperature inversions, as they would otherwise dominate figures 10 and 11 and would generally hamper the systematic analysis of weaker effects.

For clarification, we have modified the last sentence of Sect. 5.2 to

"As the impact of temperature inversions on $\delta_\Gamma$ is quite strong, we skip profiles with temperature inversions of more than 2 K for the investigation of the effects of humidity (Sect. 5.3) and surface altitude (Sect. 5.4) in order to avoid interference of different effects."

(2) use a different day in Sect. 5 for ECMWF data set than the one used for fitting.
We now moved the results for 18 December from the Supplement to Figures 3 and 7. In addition, we now also point out that the deviation of 0 for 18 June is a consequence of the fit being made for this day directly in the caption of Fig. 7.
The uncertainties given in the text already refer to the value from 18 March, which is based on a different data set than the one used for fitting.

(3) better follow the AMT guidelines regarding the usage of "quantity/ unit" notation instead of "quantity [unit]"

We appreciate the reviewers wish for consistent formatting according to journal guidelines. The format of units in figure labels is not directly specified in the current AMT guidelines. Obviously, Copernicus has to take some action here as the format used in most publications ("quantity [unit]") is not in accordance with SI brochure and green book.

We have contacted the Copernicus editorial support on this matter, and they are now aware of this inconsistency. However, as it will probably take some time to be specified by Copernicus (as this would affect all journals, not only AMT), we would like to keep the figures as they are for now.

In the case that Copernicus specifies the format in short time, we will modify the figures accordingly.

Below we respond in detail to the further issues raised by the reviewer.

Kind regards,

Steffen Beirle

The revised version of the manuscript entitled "Calculating the vertical column density of O4 from surface values of pressure, temperature and relative humidity" does address most of the reviewers comments or, where the authors disagree, they give arguments for not changing, just as they state in their answer.

In terms of structure, the manuscript improved and is now easier to follow. Adding references to specific equations and subsections in the appendix also makes it easier to follow.

In terms of validation datasets, two of the three datasets increased greatly in number of measurements compared to the original submission. I also acknowledge the extra section on temperature inversions, however, I think it is misleading to exclude temperature inversions from the analysis and urge the authors to re-include them in the analysis to reflect fairly the ability and shortcomings of the proposed method since temperature inversions cannot be detected from just measuring the surface values of temperature, pressure and relative humidity and hence their results could be unfairly biased to better values when intentionally taking out temperature inversions from the comparisons in Sect 5.

This seems to be a misunderstanding: daytime temperature inversions are not skipped from the analysis: They are included in the global maps as well as in the presented histograms and the listed standard deviations.
Only for the investigation of possible dependencies on humidity (Fig. 10) and surface altitude (Fig. 11) we had to skip temperature inversions, as they would otherwise dominate figures 10 and 11 and would generally hamper the systematic analysis of weaker effects.

For clarification, we have modified the last sentence of Sect. 5.2 to

"As the impact of temperature inversions on $\delta_\Gamma$ is quite strong, we skip profiles with temperature inversions of more than 2 K for the investigation of the effects of humidity (Sect. 5.3) and surface altitude (Sect. 5.4) in order to avoid interference of different effects."

Likewise, I urge the authors to use a different day for the ECMWF dataset in Sect.5 than the day which was used for the fit.

We now moved the results for 18 December from the Supplement to Figures 3 and 7. In addition, we now also point out that the deviation of 0 for 18 June is a consequence of the fit being made for this day directly in the caption of Fig. 7.
The uncertainties given in the text already refer to the value from 18 March, which is based on a different data set than the one used for fitting.

I mostly checked the new submission with track changes i.e. document amt-2021-213-author_response-version1.pdf. However, I see now that there are actually differences between that version and what was submitted as a new version, likely exclusively attributable to latex issues with the diff package [and hence nothing the authors could influence]. See for example page 11 (page 32) in the former document, last sentence of Sect. 3.2 "For further details see Appendix B0.1 and B0.1" and compare this with the last sentence of the document which was submitted as new version as amt-2021-213-manuscript-version2.pdf (page 9): "For further details see Appendix B1.2 and B1.3". Please ignore comments that refer to non-existing sections or multiple figure numbers if you find them correct in the not-track changed version. Line numbers below refer to the track changed version.

Yes, the incorrect labels in the tracked-changes version are caused by latexdiff.

(1) Since the authors only consider in their validation daytime VCDs (see also Sect. 3, especially lines 209- 212), the title should reflect this limitation. Please add this to the title accordingly.

We have modified the title to "Calculating the vertical column density of O4 **during daytime** from surface values of pressure, temperature and relative humidity"

(2) Regarding following the AMT guidelines:

(a) I disagree with the authors that the use of "quantity/ unit" in axes labels and table headings is only a recommendation. I have also never "noted" that it is only a recommendation either. Please read carefully the AMT author guidelines (https://www.atmospheric-measurement-techniques.net/submission.html#math):

"

In addition, the SI and IUPAC recommendations should be followed:

SI brochure

IUPAC Green Book, 3rd edition

IUPAC Gold Book

"

Collins dictionary says about the use of the word "should" (https://www.collinsdictionary.com/dictionary/english/should #2) :

"

You use should to give someone an order to do something, or to report an official order.

All visitors should register with the British Embassy.

The European Commission ruled that the company should pay back tens of millions of pounds.

"

Hence, it is more an obligation than a recommendation to follow the guidelines (or recommendations) of the SI brochure and the green/ gold book. The point is that AMT uses "should be followed" not "it is recommended to follow" or "authors are encouraged to follow".

(b) The argument that the authors give " Thus we prefer to provide units in brackets in figure axis, which is also commonly done in most of recently published AMT papers", is not a valid argument for the validity of their choice; it is merely a statement about the state of quality control at AMT regarding their own guidelines.

(c) Even if following the guidelines of the green book and the SI brochure were just a recommendation:

Who will follow these guidelines, if not even the chief editor of the journal which publishes these guidelines/ recommendations follows them? People in power should set good examples and follow (at least their own) recommendations, otherwise recommendations do not make any sense and could be removed all together.

(d) There are good reasons why AMT refers to the SI brochure and the green book. For a motivation for these guidelines, check out the preface of the green book, where it reads on page IX:

"The purpose of this manual is to improve the exchange of scientific information among the readers in different disciplines and across different nations"

Including the unit in round brackets in an axis label (or table heading) is common in some areas (e.g. physical review letters prefers the notation with round brackets is even recommended: https://journals.aps.org/prl/authors/axis-labels-and-scales-on-graphs-h18; but keep in mind that the article in question was not submitted to APS but to AMT; AMT has different style guidelines, as cited above), square brackets are in fact used to give units of a quantity as follows: [quantity] = unit, e.g.: [T] = K. Using square brackets in axis labels around units is simply not correct and should never be used. Using round brackets is accepted, but has certain disadvantages: It can be easily mistaken as a multiplication factor whereas it really should be the denominator of a quotient. Hence, labeling a distance axis as "distance (m)" could be interpreted as "distance <<times>> meters". This is incorrect. What is labeled on the ticks on the x-axis in the graph is "distance <<over>> meters" (a plain number); the notation "distance/ m" does not leave any room for interpretation and will be understood correctly independent on your field or background. A very instructive explication in German (see the original

from the BIPM linked on the AMT homepage) can be found in the German version of the SI brochure: "https://www.ptb.de/cms/fileadmin/internet/publikationen/ptb_mitteilungen/mitt2007/Heft2/PTB-Mitteilungen_2007_Heft_2.pdf", as in the original, in Sect. 5.3.1 (especially page 174 [corresponding to page 33 in the pdf]).

We appreciate the reviewers wish for consistent formatting according to journal guidelines. The format of units in figure labels is not directly specified in the current AMT guidelines. Obviously, Copernicus has to take some action here as the format used in most publications ("quantity [unit]") is not in accordance with SI brochure and green book.

We have contacted the Copernicus editorial support on this matter, and they are now aware of this inconsistency. However, as it will probably take some time to be specified by Copernicus (as this would affect all journals, not only AMT), we would like to keep the figures as they are for now.

In the case that Copernicus specifies the format in short time, we will modify the figures accordingly.

(3) Regarding the authors answer about the criticism of the limited datasets:

I acknowledge that the authors increased the number of both ECMWF and WRF and I agree that this is a sufficient coverage now. However, two comments here:

(a) regarding the GRUAN dataset, I would still like to mention that more than 70% of their ~6300 datasets come from 3 stations, and hence, I do not agree that that is a good coverage.

We agree that GRUAN does not provide good spatial coverage. However, spatial coverage is provided by the ECMWF data, while GRUAN provides very good temporal coverage for some stations.

Note that we do not claim "good coverage" for GRUAN in the manuscript.

(b) from the authors answer on page 5 from document amt-2021-213-author_response-version1.pdf, I understand that the authors submitted the first version of the manuscript in the middle of the process of creating the validation data set, anticipating the full results based on a subset of just ~15%. This is certainly bad practice and should be avoided in the future. It was highly inconsistent (2 months vs a few days) and confusing in the first version.

We will avoid this in the future.

Minor comments:

line 44: "this study" is ambiguous: does it refer to "the current study" (maybe better refer to the specific Sect. 4.1) or to "Wagner et al." (better use: "that")?

We have modified "this study" to "the current study".

line 51: To which equation does "The final equation" refer to?

We skipped the reference to "The final equation" in the introduction.

Table 1: I think "deviation to" should be "deviation from" (https://www.collinsdictionary.com/dictionary/english/deviation or https://dictionary.cambridge.org/dictionary/english/deviation) (likewise l.195)

Corrected.

line 88: Are the authors sure that they want to refer to Eq. 19 here? This seems to have nothing to do with the derivation here?

Yes, thanks, the latex labels were mixed up here. We corrected this reference to Eq. 6.

line 264: Please check the references here, it currently reads Appendix B0.1 and B0.1 There is not even any B0, Appendix B starts (as it should) with 1. (page 31, page 52 in the document.)

The wrong references are caused by latexdiff. In the plain pdf, the references are correct.

line 284/285: "correlation ... are found". "is"?

Corrected.

Figure 1: The frequency is defined as points in some "square" area made of delta-x time delta-y? Please give more detail here, otherwise the quantitative description in the color bar is not meaningful (of course, the qualitative message still comes across and I think it's a very good idea). Same for Fig. 5. , 10, 11.

We have added the following specification to the caption of Fig. 1:
"Frequency per pixel is color coded, with a binning of 100 pixels for both x and y axis, as in all 2D frequency distributions shown below."

For most of these figures: The inclusion of the colour bar in either one or all subplots is inconsistent. Consider to include it as a separate axis instead. (Although it is not wrong as it is now since multiple colour bars are included where the colour scale differs between subplots. Still it seems not very pleasing for the eye as it is now.

We would like to keep the color bars inside the figure. An additional axis on the right or bottom would shrink the actual plots, which are quite small already.

Additionally, e.g. in Fig. 10, the colour bar lacks the top axis while it s present in e.g. Fig. 11).

The top axis is present in all color bars – this might be a zoom issue of the pdf viewer?

line 311: "too low" wit respect to what? Why "too low"? They are not "too" low?

We added "as compared to a polytropic atmosphere with the same O4 VCD".

Figure 8: Caption refers to "top" and "bottom" in a figure where only left and right are present. Please include horizontal separation lines between the stations. Are the authors serious in including stations with single digit numbers (Beltsville, Darwin, Nauru, LaReunion) here?

We have corrected "top" and "bottom" to "left" and "right".

We added faint separation lines between the stations.

We see no reason to exclude stations with low number of sonde launches from this figure: of course the statistics are rather poor for these sites, which is marked by faint color as noted in the caption. However, we found it worth showing that also for these stations no unexpected behavior or larger deviations could be observed.

line 303: "agreement to"?

We would follow the recommendation of the Copernicus copy editor here.

Fig. 7: Please choose any of the other 3 days here (or best all) and not the day which was used for fitting the parameters. Using the day used for fitting does not make sense. So replace Fig.7 by Fig. C2. Especially, because your argument of "But there is also a considerable reduction of SD from 1.6% for δΓ to 1.0% for δRH" (line 366) is not that strong any longer if you actually consider days that were not used for the fit, the SD only decreases from 2.0% to 1.8% (Mar), from 1.6% to 1.3% (Sep) and from 1.3% to 1.2% for Dec.

We now moved the results for 18 December from the Supplement to Figures 3 and 7. In addition, we now also point out that the deviation of 0 for 18 June is a consequence of the fit being made for this day directly in the caption of Fig. 7. We also add the SD for 18 December to the discussion in the manuscript.

The uncertainties given in the text already refer to the value from 18 March, which is based on a different data set than the one used for fitting.

line 495: I acknowledge the footnote here, I was at first a bit puzzled here.

Yes, this was confusing on first sight.

line 513: Maybe better "equation"?

We would follow the recommendation of the Copernicus copy editor here.

line 614: "Note that the for a"? Remove "the"?

Corrected.

Figure B1: Please increase the space between the map and the x- and y- axis labels. Also: There are four "Fig. B1" (at least in the version with track changes), comment is about the first one.

Done.